# BZR1 promotes pluripotency acquisition and callus development through direct regulation of *ARF7* and *ARF19*

Thomas Ammitsøe [1,4], Elise Ebstrup [1,4], Noel Blanco-Touriñán[2], Julie Hansen[1], Christian S Hardtke [2], Morten Petersen [3✉] & Eleazar Rodriguez [1✉]

## Abstract

Plants have the remarkable ability to regenerate whole organisms through the formation of pluripotent cell masses from somatic cells. Cellular programs leading to fate change resemble lateral root (LR) formation and are chiefly regulated by auxin. Brassinosteroid signaling also plays an important role in LR formation, but little is known about the direct link between auxin and brassinosteroid components, such as BZR1 and BES1, in pluripotency acquisition. Here we show that gain-of-function mutants *bzr1-D* and *bes1-D* exhibit altered callus formation, yet disruption of these transcription factors does not cause major changes to callus formation or de novo organogenesis. Moreover, our data reveal that BZR1 displays enhanced expression in callus tissue and directly binds to the promoters of *ARF7* and *ARF19*, two master pluripotency regulators, leading to their enhanced transcription. Remarkably, callus formation is abrogated in *bzr1-D* upon disruption of these *ARFs*, emphasizing that the callus formation via BZR1 depends on these auxin signaling components. In conclusion, we depict a link between, ARF7, ARF19, and BZR1 in promoting pluripotency acquisition, portraying BZR1 as a major supporting factor in callus formation.

**Keywords** ARFs; BZR1; Callus; Pluripotency; Transcriptional Regulation
**Subject Categories** Development; Signal Transduction; Stem Cells & Regenerative Medicine

## Introduction

Plants possess extraordinary plasticity, being able to naturally dedifferentiate somatic cells back into pluripotency allowing control of differentiation and self-renewal through transcriptional, post-transcriptional, and epigenetic modifications (Li and Belmonte, 2017). The steps necessary for somatic cells to achieve pluripotency and redifferentiation for de novo organogenesis can be studied in vitro, through a process referred to as callus formation (Ikeuchi et al, 2016, 2019, 2017; Bustillo-Avendaño et al, 2018). Callus formation can be induced through tissue culture in callus-inducing media (CIM; high auxin—low cytokinin ratio) (Valvekens et al, 1988), which promotes an overdrive of hormone developmental programs to unlock pluripotency (Gordon et al, 2007; Sang et al, 2018). In particular, under CIM treatment, Xylem Pole Pericycle (XPP) cells undergo cell fate change, re-enter the cell cycle, and display increased gene expression of quiescent center (QC) markers such as *WUSCHEL-RELATED-HOMEOBOX5* (*WOX5*), *SHORTROOT* (*SHR*), and *SCARECROW* (*SCR*) (Ikeuchi et al, 2019; Sugimoto et al, 2010)—a process similar to lateral root (LR) initiation (Che et al, 2007; Iwase et al, 2017; Sugimoto et al, 2010). Both LR and callus formation are regulated by an auxin responsive signaling node composed by the repressor INDOLE-ACETIC-ACID14 (IAA14)/SOLITARY-ROOT (SLR), transcriptional activators; AUXIN RESPONSIVE FACTOR7 (ARF7) and AUXIN RESPONSIVE FACTOR19 (ARF19), and their direct downstream targets LATERAL ORGAN BOUNDARIES DOMAINS (LBDs) in such a way that auxin induces the degradation of IAA14 to unlock ARF7 and ARF19 transcriptional output (Fukaki et al, 2002; Goh et al, 2012; Okushima, et al, 2007; Okushima et al, 2005). The relevance of this signaling node to both callus and LR formation is patented by observations that the double knockout (KO) *arf7-1 arf19-2* or the gain-of-function mutant *slr* are defective in callus- and LR formation, and overexpressed *LBDs* form ectopic callus (Fan et al, 2012; Iwase et al, 2011; Lee and Seo, 2017). After callus formation, in vitro shoot regeneration can be achieved by transferring callus masses to shoot-inducing media (SIM; high cytokinin – low auxin); upon culturing callus tissue in SIM, cells undergo a novel cell fate change, from a root meristem-like structure to forming shoot apical meristems (SAM), with expression of SAM markers like *WUSCHEL* (*WUS*), *CLAVATA3* (*CLV3*) etc., which enables the formation of shoots (Gordon et al, 2007; Su et al, 2009).

While the role of auxin and cytokinin signaling during callus formation and de novo organogenesis has been well established, how other hormone signaling participates in this process is comparably less understood. For instance, the steroid hormones brassinosteroids (BR) are known to be important for the regulation of a range of

[1]Section for Functional Genomics, Department of Biology, University of Copenhagen, 2200 Copenhagen, Denmark. [2]Department of Plant Molecular Biology, University of Lausanne, CH-1015 Lausanne, Switzerland. [3]Department of Plant and Environmental Sciences, University of Copenhagen, 1871 Frederiksberg, Denmark. [4]These authors contributed equally: Thomas Ammitsøe, Elise Ebstrup. ✉E-mail: morten.petersen@plen.ku.dk; eleazar.rodriguez@bio.ku.dk

developmental aspects both above and below ground (e.g. (Oh et al, 2014; Bajguz et al, 2020; Hacham et al, 2011; Planas-Riverola et al, 2019; Saito et al, 2018; Lee et al, 2019)), among them LR formation (Bao et al, 2004; Rovere et al, 2022). BR signaling in the root is mainly required for appropriate cell elongation and cellular anisotropy, and for restricting the extent of formative divisions (Fridman et al, 2021; Graeff et al, 2021, Kang et al, 2017) and has been suggested to contribute to properly integrated sieve element differentiation (Graeff et al, 2020). BR is percieved by the leucine-rich-repeat receptor kinase, BRASSI-NOSTEROID INSENSITIVE 1 (BRI1) which leads to the inactivation of the shaggy-like gene kinase BRASSINOSTEROID INSENSITIVE 2 (BIN2) and the dephosphorylation of the BES1/BZR1 BR transcription factor (TF) family, consisting of BRASSINAZOLE RESISTANT 1 (BZR1) and BRI1-EMS-SUPPRESSOR 1 (BES1) and the four BES1/BZR1 Homologs (BEH1-4), which regulates BR responsive genes in the nucleus (Kim et al, 2024, 2009; Manghwar et al, 2022; Ryu et al, 2007; Wang et al, 2002; Yin et al, 2002). The importance of the phosphorylation status to BZR1 and BES1 function have been established by the isolation of dominant mutant alleles of both TFs with a point mutation, preventing BIN2 mediated phosphorylation and consequent turnover of BZR1 and BES1 (Yin et al, 2002; Wang et al, 2002). Interestingly, while BZR1 and BES1 share 88% homology in sequence, contain similar domains, (He et al, 2005; Nolan et al, 2020; Sun et al, 2010; Yu et al, 2011) and bind to E-box (CANNTG) and BRRE sites in the promoter of target genes, the dominant mutants bzr1-D and bes1-D can exhibit different phenotypes—suggesting two different biological functions concerning plant development (Wang et al, 2002; Yin et al, 2002). Further research proposed that BES1 activates BR biosynthesis and signaling involved in plant growth (Yu et al, 2011), whereas BZR1 could function as a transcriptional repressor, binding directly to promoters of BR components to reduce BR levels (He et al, 2005). In addition to having a role as a transcriptional repressor, BZR1 has also been linked to the promotion of growth, downstream of BR components (He et al, 2005). Contrarily, BZR1 has been suggested to inhibit organ boundary formation, an important process in plant organogenesis, through the repression of CUP-SHAPED COTYLEDON (CUC) genes in the primordium cells located in the shoot apical meristem (Gendron et al, 2012). Plants exhibiting enhanced BR responses, such as bes1-D, display a decrease in procambial cell layers within the hypocotyl, indicating BES1-dependent promotion of xylem differentiation; notably, this was not seen in bzr1-D (Kondo et al, 2014). The bes1-D mutant has also been shown to negatively affect meristem size and growth due to a premature exit of the cell cycle in the early differentiation of columella stem cells (CSC) (González-García et al, 2011). Additionally, BES1 has been shown to control key processes of cell division and cell elongation, for instance, promoting differentiation through WOX5 expression in QC cells and CSCs (González et al, 2011; Lee et al, 2015) and through repressing of BRAVO activity in the QC zone in the root (Vilarrasa-Blasi et al, 2014; Lozano-Elena et al, 2018).

The fact that auxin typically serves as the primary driving force in root development highlights an interconnected link between BR and auxin. These two hormones are known to share several target genes and have overlapping activities notably in the promotion of LR formation (Bao et al, 2004; Chaiwanon and Wang, 2015; Kim et al, 2007; Li et al, 2005; Mazzoni-putman et al, 2021). Auxin has been shown to promote BZR1's nuclear accumulation and direct binding and transcriptional control of ARF7 during hypocotyl elongation (Yu et al, 2023), thus establishing a direct molecular link between BZR1

and auxin signaling pathway (Zhou et al, 2013). Similarly, BIN2 has been shown to be a positive regulator of LR organogenesis (Cho et al, 2014) and callus formation (Lee and Seo, 2017) through direct regulation of ARF7 and ARF19. Moreover, a recent study has shown BR to both promote auxin levels and repress the transcriptional output of auxin signaling in established meristems as well as de novo-formed meristems (Ackerman-Lavert et al, 2021). Although the interplay between auxin and BR in root development has been linked, the interchange and underlying mechanisms are still not fully understood.

In this study, we investigate the involvement of BZR1 and BES1 transcription factors in the processes of callus formation and de novo organogenesis. Specifically, we discovered that BZR1 expression is enhanced in root explants cultured in CIM. Consistent with this, plants expressing the dominant form of BZR1, bzr1-D, exhibit amplified callus masses and an increase in the abundance of ARF7 and ARF19 transcripts. Furthermore, we identified that BZR1 directly targets the promoters of ARF7 and ARF19, which are crucial for the dedifferentiation of somatic cells into pluripotent cells in bzr1-D, revealing an important contribution of brassinosteroid signaling components to callus formation.

# Results

## Stabilization of BZR1 and BES1 have different outcomes during pluripotency acquisition and de novo shoot formation

Given the interesting proposition of a direct link between auxin and different BR signaling components like BZR1 during developmental changes (e.g., hypocotyl elongation), we wondered if this could also be observed in bzr1-D and bes1-D during pluripotency acquisition and callus formation. To test this, we cultivated root explants of bzr1-D and bes1-D on CIM media for 21 days and evaluated their performance during callus formation (Fig. 1A). Surprisingly, bzr1-D exhibited a significant increase in callus mass compared to the Col-0 wild-type (WT), while bes1-D produced slightly lower callus mass compared to WT (Fig. 1A,B).

Next, we wanted to explore if the differences in callus output between bes1-D and bzr1-D were also observable during de novo shoot formation after CIM treatment. To examine this, we transferred 21-day-old CIM-grown explants to SIM and followed them for another 21 days (Fig. 1A). Interestingly, culturing bes1-D on SIM leads to improvement in callus formation, being able to catch up to, and exceed, WT mass (Fig. 1C). We also noticed the shoot formation was significantly higher in bes1-D compared to WT (Fig. 1D). As for bzr1-D, explants from this genotype still displayed a significantly higher amount of callus mass than the WT after SIM incubation. Similar to bes1-D, bzr1-D also displayed a higher number of shoots compared to WT (Fig. 1D). BR has previously been associated with the repression of organ boundary formation important for the shoot apical meristem (Gendron et al, 2012), thus in combination, these results suggest that stabilization and expansion of BES1/BZR1 facilitates and enlarges shoot meristematic area.

## BZR1 and BES1 are not crucial for callus and shoot growth

Intrigued by our results regarding the dominant versions of BZR1 and BES1, we next wanted to examine the callus-related phenotypes of the

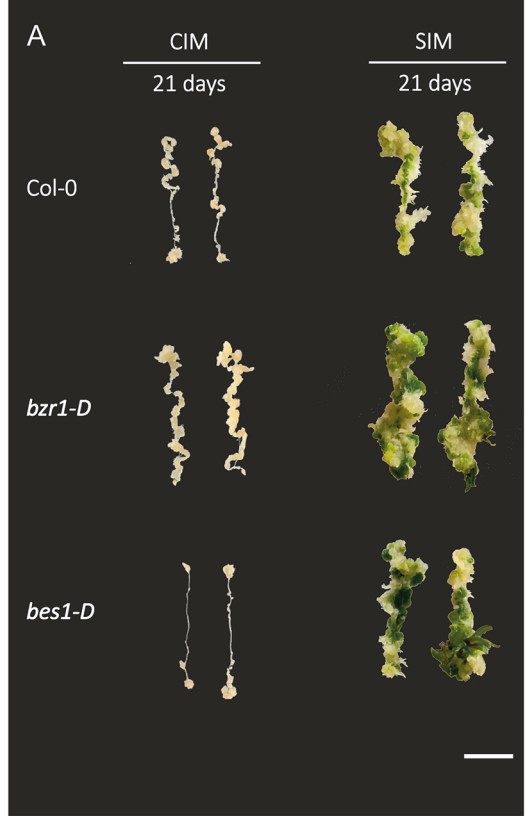

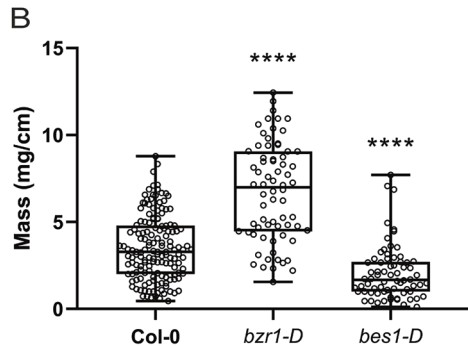

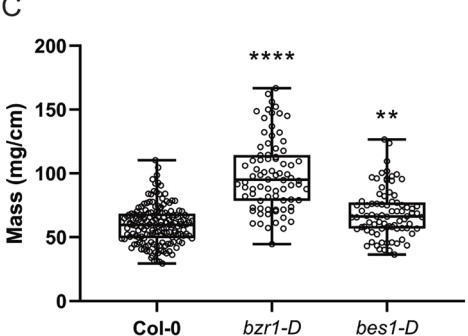

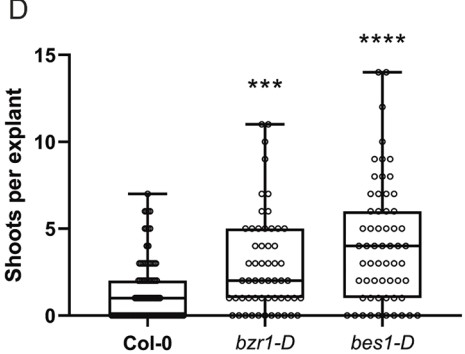

**Figure 1. Stabilization of BES1 and BZR1 promotes different outcomes during pluripotency acquisition and shoot formation.**

(A) Representative images of callus forming root explants Col-0, *bzr1-D*, and *bes1-D* from 21-day CIM or 21-day SIM (21-day CIM + 21-day SIM); scale bar: 1 cm. (B) Fresh weight of 21-day root explants on CIM (Col-0 *n* = 152, *bzr1-D* *n* = 70, *bes1-D* *n* = 70). (C) Fresh weight of 21-day-explants on SIM. (D) Shoots counted per callus explant after 21 days on SIM. Data information: (C, D) (Col-0 *n* = 159, *bzr1-D* *n* = 80, *bes1-D* *n* = 82). Experiments were performed four times with similar results. Significance was determined using the Kruskal–Wallis nonparametric test (*P* < 0.05), followed by Dunn's multiple comparisons test with Bonferroni adjustment. Asterisks indicate significant differences from Col-0 (**$P$ < 0.01, ***$P$ < 0.001, ****$P$ < 0.0001). Comparisons were made between Col-0, *bzr1-D*, and *bes1-D*. (B) Significant differences were observed between Col-0 vs *bzr1-D* ($P$ = 5.0 × 10$^{-324}$) and between Col-0 vs *bes1-D* ($P$ = 4.27 × 10$^{-11}$). (C) A significant difference was observed between Col-0 vs *bzr1-D* ($P$ = 5.0 × 10$^{-324}$), and between Col-0 vs *bes1-D* ($P$ = 0.0035). (D) Significant differences were observed between Col-0 and *bzr1-D* ($P$ = 0.0012) and between Col-0 vs *bes1-D* ($P$ = 1.16 × 10$^{-7}$). Boxplots: Centerlines indicate the median; box limits represent the 25th and 75th percentiles; whiskers extend to the minimum and maximum values. Source data are available online for this figure.

loss-of-function mutants of these genes. A *BES1* T-DNA line was available from the Arabidopsis seed stock center (Alonso et al, 2003) but none was available for *BZR1* and we thus generated them by Crispr–CAS9 gene editing. We isolated two knockout lines of *BZR1*, *bzr1-c1* and *bzr1-c2*, one with a 1 bp insertion and the other with a deletion of a single nucleotide (at position 144 and 143 of the *BZR1*

*cds*) respectively, both resulting in a frameshift mutation disrupting the gene (Fig. EV1). The three mutants were cultivated on CIM for a period of 21 days, as illustrated in Fig. 2A. Interestingly, *bes1-ko* and *bzr1-c1* showed no significant difference, whereas *bzr1-c2* displayed increased mass, compared to the WT, however no statistical difference was found between the two *bzr1-c* lines (Fig. 2B). This indicates that

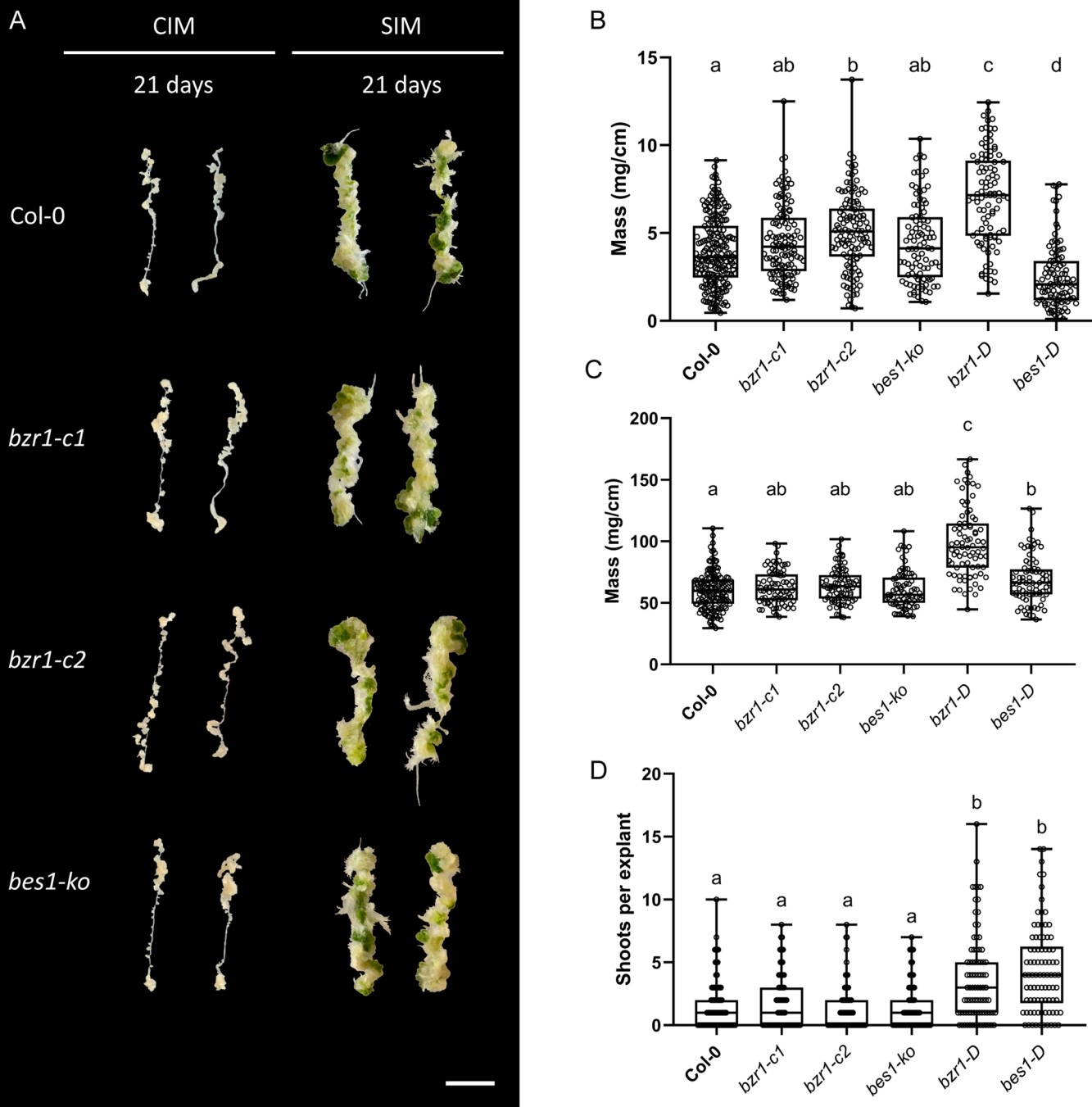

the absence of these two transcription factors does not negatively impact callus formation. As in Fig. 1, we examined the effect on the mutants when transitioning from CIM to SIM, and in line with the results depicted in Fig. 2B, no significant differences were found between the mutants and WT in terms of de novo organogenesis (Fig. 2C,D). These findings indicate that the absence of BZR1 and BES1 does not produce a significant effect on callus or de novo shoot formation, which could be explained by the known redundancy of the BES1/BZR1 TF family (Chen et al, 2019). In support of this proposition, we observed that the expression of all *BEHs* was gradually

induced in WT explants by CIM treatment when compared to control explants (Fig. EV2).

## *bzr1-D* and *bes1-D* act independently of BR synthesis but shows contrasting involvement of BIN2

Given our results above, we wanted to further investigate to what extent the observed callus phenotype of *bzr1-D* and *bes1-D* relies on BR signaling. Callus formation analysis of brassinosteroid signaling and biosynthetic pathway mutants like *det2, bri1 and bzr*-sextuplet

◄ **Figure 2. BZR1 and BES1 are dispensable for callus formation.**

(A) Representative images of callus forming root explants Col-0, *bzr1-c1*, *bzr1-c2*, and *bes1-KO* from 21-day CIM or 21-day SIM (21-day CIM + 21-day SIM); scale bar: 1 cm. (B) Fresh weight of 21-day root explants on CIM (Col-0 $n = 214$, *bzr1-c1* $n = 119$, *bzr1-c2* $n = 119$, *bes1-ko* $n = 92$, *bzr1-D* $n = 96$, *bes1-D* $n = 96$). (C) Fresh weight of 21-day explants on SIM. (D) Shoots counted per callus explant after 21 days on SIM. Data information: (C, B) (Col-0 $n = 159$, *bzr1-c1* $n = 79$, *bzr1-c2* $n = 81$, *bes1-ko* $n = 75$, *bzr1-D* $n = 80$, *bes1-D* $n = 82$). Experiments were performed 5 times with similar results. Kruskal–Wallis test followed by Dunn's multiple comparisons test ($P < 0.05$), with Bonferroni adjustment, was performed to compare Col-0, *bzr1-c1*, *bzr1-c2*, *bes1-ko*, *bzr1-D*, and *bes1-D*. Significant differences were observed between Col-0 vs *bzr1-c2* ($P = 2.96 \times 10^{-4}$), Col-0 vs *bzr1-D* ($P = 5.0 \times 10^{-324}$), Col-0 vs *bes1-D* ($P = 1.19 \times 10^{-6}$), *bzr1-c1* vs *bzr1-D* ($P = 9.28 \times 10^{-9}$), *bzr1-c1* vs *bes1-D* ($P = 7.49 \times 10^{-10}$), *bzr1-c2* vs *bzr1-D* ($P = 1.55 \times 10^{-4}$), *bzr1-c2* vs *bes1-D* ($P = 5.0 \times 10^{-324}$), *bes1-ko* vs *bzr1-D* ($P = 1.55 \times 10^{-8}$), *bes1-ko* vs *bes1-D* ($P = 5.29 \times 10^{-8}$), *bzr1-D* vs *bes1-D* ($P = 5.0 \times 10^{-324}$). Comparisons that did not reach statistical significance included Col-0 vs. *bzr1-c1* ($P = 0.5112$), Col-0 vs. *bes1-ko* ($P = 1.00$), *bzr1-c1* vs. *bzr1-c2* ($P = 0.8875$), *bzr1-c1* vs. *bes1-ko* ($P = 1.00$), and *bzr1-c2* vs. *bes1-ko* ($P = 0.5941$). For (C), significant differences were observed between Col-0 vs *bzr1-D* ($P = 5.0 \times 10^{-324}$), Col-0 vs *bes1-D* ($P = 1.24 \times 10^{-2}$), *bzr1-c1* vs *bzr1-D* ($P = 3.33 \times 10^{-15}$), *bzr1-c2* vs *bzr1-D* ($P = 9.66 \times 10^{-14}$), *bes1-ko* vs *bzr1-D* ($P = 5.0 \times 10^{-324}$), and *bzr1-D* vs *bes1-D* ($P = 2.08 \times 10^{-9}$). Comparisons that did not reach statistical significance included Col-0 vs. *bzr1-c1* ($P = 1.00$), Col-0 vs. *bzr1-c2* ($P = 1.00$), Col-0 vs. *bes1-ko* ($P = 1.00$), *bzr1-c1* vs. *bzr1-c2* ($P = 1.00$), *bzr1-c1* vs. *bes1-ko* ($P = 1.00$), *bzr1-c1* vs. *bes1-D* ($P = 0.8316$), *bzr1-c2* vs. *bes1-ko* ($P = 1.00$), and *bzr1-c2* vs. *bes1-D* ($P = 1.00$). For (D), significant differences were observed between Col-0 vs *bzr1-D* ($P = 1.36 \times 10^{-6}$), Col-0 vs *bes1-D* ($P = 2.69 \times 10^{-11}$), *bzr1-c1* vs *bzr1-D* ($P = 2.59 \times 10^{-4}$), *bzr1-c1* vs *bes1-D* ($P = 1.30 \times 10^{-7}$), *bzr1-c2* vs *bzr1-D* ($P = 4.86 \times 10^{-7}$), *bzr1-c2* vs *bes1-D* ($P = 3.70 \times 10^{-11}$), *bes1-ko* vs *bzr1-D* ($P = 1.47 \times 10^{-5}$), and *bes1-ko* vs *bes1-D* ($P = 3.60 \times 10^{-9}$). Comparisons that did not reach statistical significance included Col-0 vs. *bzr1-c1* ($P = 1.00$), Col-0 vs. *bzr1-c2* ($P = 1.00$), Col-0 vs. *bes1-ko* ($P = 1.00$), *bzr1-c1* vs. *bzr1-c2* ($P = 1.00$), *bzr1-c1* vs. *bes1-ko* ($P = 1.00$), *bzr1-c2* vs. *bes1-ko* ($P = 1.00$), and *bzr1-D* vs. *bes1-D* ($P = 1.00$). Boxplots: Centerlines indicate the median; box limits represent the 25th and 75th percentiles; whiskers extend to the minimum and maximum. Source data are available online for this figure.

would be complicated by their reduced root size and pleiotropism (Li and Chory, 1997; Park et al, 2014). To circumvent this issue, we decided to treat root explants grown on CIM with either Brassinazole (BRZ), a BR biosynthesis inhibitor, or the GSK3 inhibitor Bikinin (BIK), thus inactivating BIN2. This way, we could analyze the effect of BR synthesis and signaling inhibition specifically in the context of callus formation rather than observing chronic effects of BR-impairment from germination.

The effect of BR synthesis inhibition was evident in WT, where BRZ greatly reduced callus mass compared to the DMSO mock (Fig. 3A). In contrast both *bzr1-D* and *bes1-D* were not significantly affected by BRZ treatment (Fig. 3A,B). These results both indicate that BR synthesis has a positive effect on callus formation and that the *bzr1-D* and *bes1-D* mutants can bypass BR synthesis limitations, which is in line with previous findings regarding BZR insensitivity of *BES1* and *BZR1* gain-of-function mutants (Wang et al, 2002; Yin et al, 2002).

Concerning BIK treatment, we observed a marked reduction in callus mass for both WT and *bzr1-D*, but surprisingly not *bes1-D* (Fig. 3A). Importantly, BIK-treated *bzr1-D* callus mass was comparable to that of mock-treated WT and significantly heavier than those of BIK-treated WT (Fig. 3A,B). This result suggests that while callus formation in *bzr1-D* is still sensitive to BIN2 signaling, this effect is likely additive. In contrast to *bzr1-D*, *bes1-D* seems to function largely autonomous to BIN2 signaling, which might not be surprising given that BES1 has been shown to be dephosphorylated and activated in a BR-independent manner (Albertos et al, 2022). This result is also in line with reports by (Cho et al, 2014; Lee and Seo, 2017) showing that BIN2 promotes callus formation in a BR-independent manner.

## CIM and SIM promote different fates for BES1 and BZR1 expression

To investigate how callus formation affects the expression patterns of *BZR1* and *BES1*, we utilized transgenic lines expressing the translational reporter *pBES1::BES1-YFP* (BES-YFP) or *pBZR1::BZR1-YFP* (BZR1-YFP) (Chen et al, 2019). We examined their fluorescence intensity by confocal microscopy upon culturing in CIM at the root tip, in the XPP (NT treatment) and in the callus midlayer (CIM treatment), which has been shown to be the region

for pluripotency acquisition in callus explants (Zhai and Xu, 2021). For this purpose, 7-day-old root explants were treated with either 6 days in liquid CIM or CIM without hormones, with solvents (NT). In mature tissue, only low levels of BES1-YFP signal were detected in NT conditions, but this was further suppressed under CIM treatment (Fig. 4A,B). As for BZR1, CIM treatment increased BZR1-YFP signal in mature tissue, however this restricted to the callus midlayer (Fig. 4A,C). As for the root tip, BES1-YFP displayed nuclear localization in the root tip under control conditions (Fig. 4D); this signal was decreased significantly upon CIM treatment (Fig. 4D,E). BZR1-YFP also showed nuclear localization at the root tip in NT conditions (Fig. 4B). However, and in marked contrast to BES1-YFP, the signal of BZR1-YFP increased substantially in the root tip following CIM treatment (Fig. 4D,F). Altogether, these results showing opposite outcomes for BES1 and BZR1 expression under CIM treatment are in line with the callus phenotypes observed in Fig. 1. The increased BZR1 signal could stem from the high levels of auxin in the CIM treatment, as a previous study has shown induced nuclear localization of BZR1 in the hypocotyl, under auxin-induced elongation (Yu et al, 2023). In addition, auxin and BR share overlapping activities in the promotion of LR (Bao et al, 2004; Chaiwanon and Wang, 2015; Kim et al, 2007; Li et al, 2005; Mazzoni-putman et al, 2021), suggesting a possible positive role for BZR1 in upregulating targets important for callus formation. Interestingly, BZR1 showed strong nuclear localization in the callus midlayer which shows a QC-like transcriptional profile (Zhai and Xu, 2021), suggesting a role of BZR1 in promoting pluripotency specifically in the callus midlayer.

Given the vast difference in *BES1* and *BZR1* expression in CIM media and its correlation with the callus phenotypes of *bes1-D* and *bzr1-D*, we wanted to assess if a similar trend would be observed during the transition to SIM. We addressed this by transferring explants to SIM after a 21-day CIM treatment. In these conditions, and contrasting with what was observed for CIM, BES1-YFP signal increased dramatically within just 3 days of treatment with SIM (Fig. 4A,B). This is consistent with the phenotype displayed by *bes1-D* under SIM treatment, namely increased shoot formation (Fig. 1). As for BZR1-YFP, the expression of this marker was reminiscent of that displaying CIM, i.e. high expression throughout the entire callus mass (Fig. 4A,C).

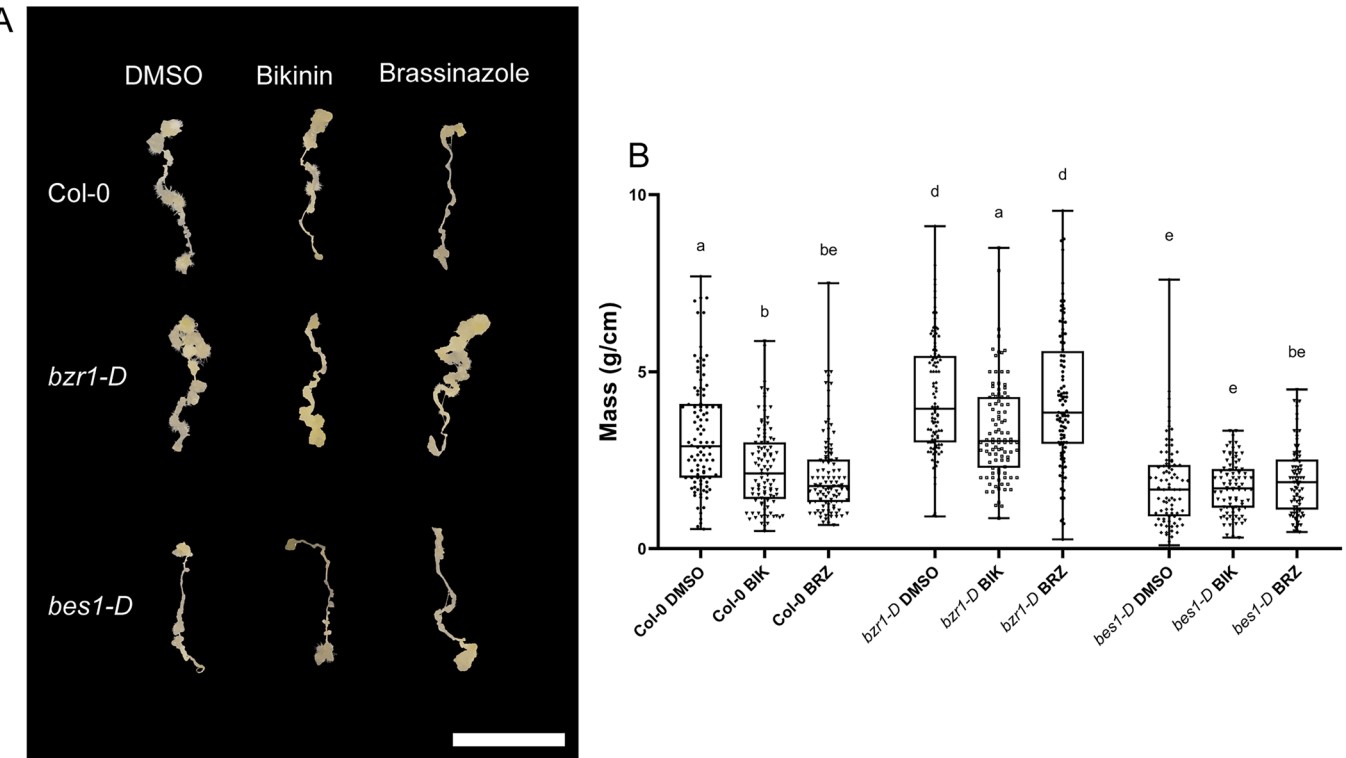

**Figure 3. *bzr1-D* and *bes1-D* display a BR-independent role during callus formation.**

Callus forming root explants of Col-0, *bzr1-D*, *bes1-D* from 21-day CIM DMSO (mock), 5 µM Bikinin (*BIK*) or 2 µM Brassinazole (*BRZ*). (**A**) Representative images of callus from 21-day CIM + inhibitors; scale bar: 1 cm. (**B**) Fresh weight of 21-day root explants on CIM after treatment with inhibitors (Col-0: DMSO $n = 100$, BIK $n = 99$, BRZ $n = 101$. *bzr1-D*: DMSO $n = 100$, BIK $n = 100$, BRZ $n = 100$. *bes1-D*: DMSO $n = 83$, BIK $n = 87$, BRZ $n = 85$). The scale bar corresponds to 1 cm. Data information: Experiments were performed three times with similar results. Kruskal–Wallis test followed by Dunn's multiple comparisons test ($P < 0.05$), with Bonferroni adjustment, was performed to compare Col-0, *bzr1-D*, and *bes1-D* under different treatments (*DMSO*, *BIK*, and *BRZ*). Significant differences were observed between Col-0 *DMSO* vs *bzr1-D DMSO* ($P = 1.72 \times 10^{-6}$), Col-0 *DMSO* vs *bes1-D DMSO* ($P = 5.0 \times 10^{-324}$), Col-0 *DMSO* vs Col-0 *BIK* ($P = 7.04 \times 10^{-6}$), Col-0 *DMSO* vs Col-0 *BRZ* ($P = 2.29 \times 10^{-11}$), Col-0 *DMSO* vs *bzr1-D BRZ* ($P = 2.54 \times 10^{-4}$), Col-0 *DMSO* vs *bes1-D BRZ* ($P = 5.44 \times 10^{-12}$), *bzr1-D DMSO* vs *bes1-D DMSO* ($P = 5.0 \times 10^{-324}$), *bzr1-D DMSO* vs Col-0 *BIK* ($P = 5.0 \times 10^{-324}$), *bzr1-D DMSO* vs *bzr1-D BIK* ($P = 1.25 \times 10^{-3}$), *bzr1-D DMSO* vs *bes1-D BIK* ($P = 5.0 \times 10^{-324}$), *bzr1-D DMSO* vs Col-0 *BRZ* ($P = 5.0 \times 10^{-324}$), *bzr1-D DMSO* vs *bzr1-D BRZ* ($P = 5.0 \times 10^{-324}$), vs *bzr1-D DMSO* vs *bes1-D BRZ* ($P = 5.0 \times 10^{-324}$), *bes1-D DMSO* vs Col-0 *BIK* ($P = 2.22 \times 10^{-2}$), *bes1-D DMSO* vs *bzr1-D BIK* ($P = 2.61 \times 10^{-9}$), *bes1-D DMSO* vs *bes1-D BIK* ($P = 8.80 \times 10^{-3}$), Col-0 *BIK* vs *bzr1-D BIK* ($P = 2.61 \times 10^{-9}$), Col-0 *BIK* vs *bes1-D BIK* ($P = 8.80 \times 10^{-3}$), Col-0 *BIK* vs *bzr1-D BRZ* ($P = 5.0 \times 10^{-324}$), Col-0 *BIK* vs *bes1-D BRZ* ($P = 5.0 \times 10^{-324}$), *bzr1-D BIK* vs *bes1-D BIK* ($P = 5.0 \times 10^{-324}$), *bzr1-D BIK* vs Col-0 *BRZ* ($P = 5.0 \times 10^{-324}$), *bzr1-D BIK* vs *bes1-D BRZ* ($P = 5.0 \times 10^{-324}$), Col-0 *BRZ* vs *bzr1-D BRZ* ($P = 5.0 \times 10^{-324}$), *bzr1-D BRZ* vs *bes1-D BRZ* ($P = 5.0 \times 10^{-324}$). Comparisons that did not reach statistical significance included Col-0 *DMSO* vs *bzr1-D BIK* ($P = 1.00$), *bzr1-D DMSO* vs *bzr1-D BRZ* ($P = 1.00$), *bes1-D DMSO* vs *bes1-D BIK* ($P = 1.00$), *bes1-D DMSO* vs Col-0 *BRZ* ($P = 1.00$), *bes1-D DMSO* vs *bes1-D BRZ* ($P = 1.00$), Col-0 *BIK* vs Col-0 *BRZ* ($P = 1.00$), Col-0 *BIK* vs *bes1-D BRZ* ($P = 0.6211$), *bzr1-D BIK* vs *bzr1-D BRZ* ($P = 0.0543$), *bzr1-D BIK* vs Col-0 *BRZ* ($P = 1.00$), *bes1-D BIK* vs *bzr1-D BRZ* ($P = 1.00$), *bes1-D BIK* vs *bes1-D BRZ* ($P = 1.00$). Boxplots: Centerlines indicate the median; box limits represent the 25th and 75th percentiles; whiskers extend to the minimum and maximum. Source data are available online for this figure.

Altogether, our data suggest that during CIM treatment, stabilization of BZR1-YFP and consequent increased nuclear localization facilitates pluripotency acquisition and cellular proliferation, leading to increased callus production, while BES1-YFP shows the opposite response. However, both transcription factors are induced upon SIM culturing, and given the enhanced shoot formation seen for *bes1-d* and *bzr1-d*, altogether suggests that both promote shoot formation.

## BZR1 binds and regulates *ARF7* and *ARF19* under callus formation

Our combined results showing enhanced *BZR1* expression on CIM and increased callus mass of *bzr1-D* suggest that BZR1 might have a role in mediating pluripotency acquisition. To guide our search for direct targets of BZR1, we performed chromatin

immunoprecipitation sequencing (ChIP-seq) on callus cultured on CIM for 21 days. We utilized the BZR1-YFP line against a *pUBQ10::YFP-GOT1* (YFP-GOT1) control and identified several targets of BZR1. We further selected some of those targets, based on their known function in stem cell formation and regulation and verified them by ChIP. Precisely, we found BZR1 to bind the QC-specific transcription factor *SCR* (Tian et al, 2022), which is a known target of BZR1. Furthermore, we identified a novel target of BZR1, namely the small secreted peptide *Root Meristem Growth Factor 1* (*RGF1*), which is required for stem cell niche maintenance through regulation of PLT2 (Fig. EV3A,B). (Matsuzaki et al, 2010; Yamada et al, 2020) Interestingly, we observed that BZR1 binds to the promoter regions of the pluripotency master regulator *ARF7*, which had been identified previously as a target of BZR1 during hypocotyl elongation (Zhou et al, 2013). Importantly, and to our

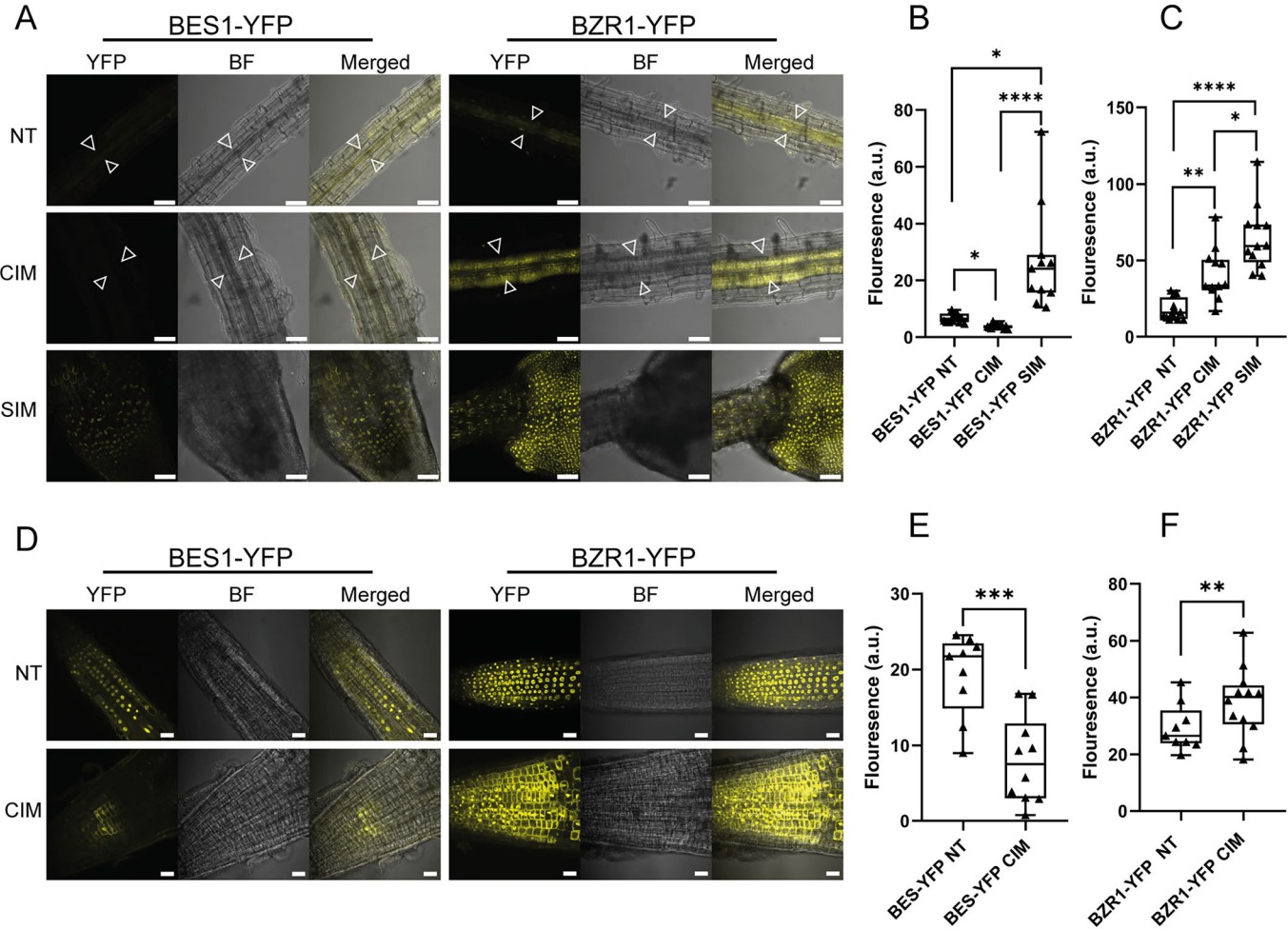

**Figure 4. BES1 and BZR1 expression during callus formation and de novo shoot formation.**

(A) Confocal images of BES1-YFP and BZR1-YFP 7-day-old root explants followed with 6-day treatment of liquid CIM or NT (CIM without hormones, with solvents) in Col-0 background. White arrows mark the stele/callus midlayer. For SIM treatment, 21-day CIM explants were transferred to SIM for 3 days. Scale bars correspond to 50 µm. (B, C) Quantification of mean gray value of BES1-YFP and BZR1-YFP ($n = 12$ per line per condition) from (A). (D) Confocal images of root tips of BES1-YFP and BZR1-YFP 7-day-old root explants followed with 6-day treatment of liquid CIM or NT (CIM without hormones, with solvents) in Col-0 background. Scale bars corresponding to 20 µm. Data information: (B) Kruskal–Wallis test followed by Dunn's multiple comparisons test ($P < 0.05$), with Bonferroni adjustment, was performed to compare NT, CIM, and SIM treatments. BES1-YFP NT vs CIM ($P = 0.0202$) NT vs SIM (*$P = 0,017$) CIM vs SIM (****$P = 7.08 \times 10^{-8}$). (C) Statistical analysis was performed using a one-way ANOVA with Tukey's test for multiple comparisons ($P < 0.05$). (C) BZR1-YFP NT vs CIM ($P = 0.001457$) NT vs SIM (****$P < 0.0001$) CIM vs SIM (*$P = 0.0439$). (E, F) Quantification of the mean gray value of BES1-YFP and BZR1-YFP from (D) (E BES1-YFP RT: NT $n = 9$, CIM $n = 10$; F BZR1-YFP RT: NT $n = 9$, CIM $n = 12$). Statistical analysis was performed using Student's unpaired $t$ test. (E) BES1-YFP RT NT vs CIM (***$P = 0.0004$). (F) BZR1-YFP RT NT vs CIM (**$P = 0.0011$) Boxplots: centerlines show the median; the box limits the 25th and 75th percentiles; whiskers extend to the minimum and maximum. Asterisks indicate significant differences from Col-0 (*$P < 0.05$, **$P < 0.01$, ***$P < 0.001$, ****$P < 0.0001$). Source data are available online for this figure.

knowledge for the first time, we also found that *ARF19*, which works redundantly with *ARF7* in pluripotency acquisition (Fan et al, 2012), is a target of BZR1. Due to this, we decided to examine the relative expression levels of both *ARFs*, which consistent with our reasoning, was significantly elevated upon 21 days of CIM treatment in *bzr1-D* when compared to WT (Fig. 5A,B). Notably, the expression *ARF7* and *ARF19* in *bes1-D* was not different from WT levels during CIM treatment (Fig. EV4A,B), suggesting *bes1-D* does not regulate *ARF7*/19 at a transcriptional level during callus formation. Encouraged by these results, we confirmed whether BZR1 binds *ARF7* and *ARF19* promoter regions during callus induction through ChIP. As indicated in Fig. 5C, the promoter

regions and initial gene segments of both *ARF7* and *ARF19* exhibit the presence of E-box and BRRE elements. We chose to test DNA binding of BZR1 on two predicted promoter region binding sites at position −268, +543, and −2348, −982 from the transcriptional start site for *ARF7* and *ARF19*, respectively. Following the 21-day CIM treatment, we observed enriched binding of BZR1 to the selected promoter regions of *ARF7* and *ARF19* specifically (Figs. 5D and EV4C). These results expand on the known role of BZR1's regulation of plant development via *ARF7* (Zhou et al, 2013), now adding callus formation to this repertoire. Moreover, we identified *ARF19* as a novel target of BZR1 (Fig. 5) during callus formation, further expanding the complex role of *BZR1* during

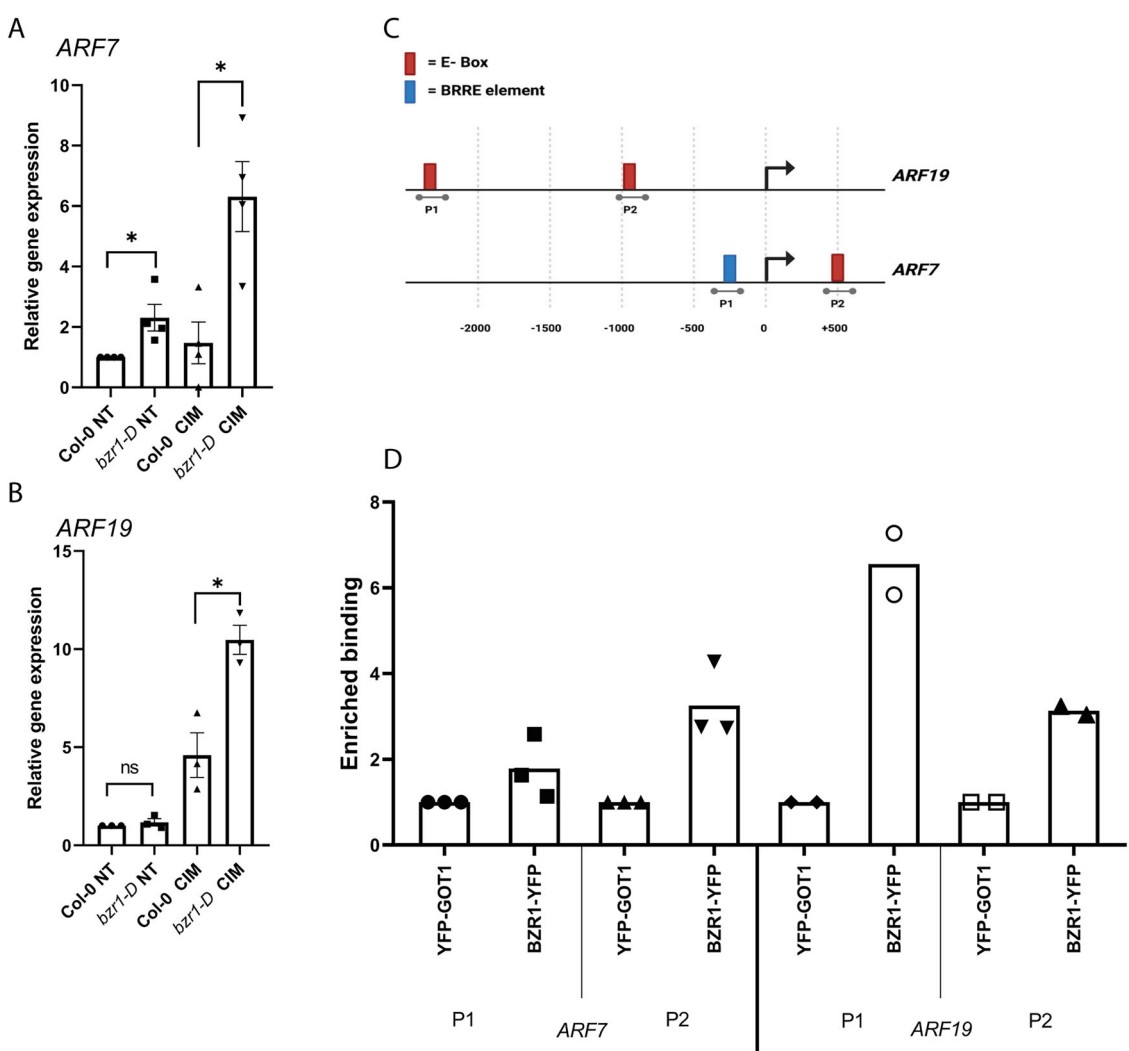

**Figure 5. Callus induction enhances BZR1 binding to the promoters *ARF7* and *ARF19*.**

(A) Relative expression of *ARF7* and (B) *ARF19* were analyzed in Col-0 and *bzr1-D*. Tissue was obtained from NT and 21-day CIM-treated root explants. Expression was normalized to *ACTIN2* and is relative to Col-0 NT; i.e. root explants from 7-day-old seedlings grown on MS media. Error bars indicate the SEM of the mean. Significance was calculated via unpaired Student's *t* test (*P* < 0.05). Asterisks represent the significant difference from Col-0 NT/CIM from each treatment (**P* < 0.05). (A) NT (**P* = 0.0249) CIM (**P* = 0.0115). (B) CIM (**P* = 0.0124). Experiments were performed in *n* = 4 (*ARF7*) and 3 (*ARF19*) biological replicates, each with three technical replicates. Each replicates consisting of 100 mg root tissue (NT) or 16 callus explants (CIM). (C) Schematic overview of promoter regions for *ARF7* and *ARF19*. E-box and BRRE-element are marked accordingly, and primer sets are boarding the sites. (D) ChIP experiment showing the DNA binding of BZR1-YFP to *ARF7* and *ARF19* promoter regions. This was determined through quantitative RT-PCR. Fold change was calculated by normalizing each sample against its input and is presented relative to the YFP-GOT1 control for each sample. Bars show the mean of *n* = 3 biological replicates for *ARF7* and *n* = 2 biological replicates for *ARF19*. Source data are available online for this figure.

developmental regulation (Chaiwanon and Wang, 2015; Gendron et al, 2012). Together these findings demonstrate that the binding of BZR1 leads to an increased expression of *ARF7* and *ARF19*, which potentially facilitates callus formation.

## *bzr1-D* enhanced callus formation is dependent on *ARF7* and *ARF19*

In light of our results showing that BZR1 directly regulates *ARF7* and *ARF19* during CIM treatment, we wondered if the enhanced callus formation phenotype of *bzr1-D* was dependent on these two

ARFs. To test this, we generated a triple mutant line, *bzr1-D arf7-1 arf19-2* (Fig. 6), and analyzed its callus formation ability. Consistent with our hypothesis, the *bzr1-D*-positive effect on callus formation was abolished in the *arf7-1 arf19-2* background (Fig. 6A,B).

As LR initiation and callus formation share a developmental program, we wanted to evaluate LR phenotype of *bzr1-D* and its potential dependency on *ARF7* and *ARF19*. Interestingly and as seen in Fig. 6C,D, *bzr1-D* produced significantly more LRs than WT, which was yet unreported for this mutant and is in contrast to previous results (Cho et al, 2014). In agreement with our callus

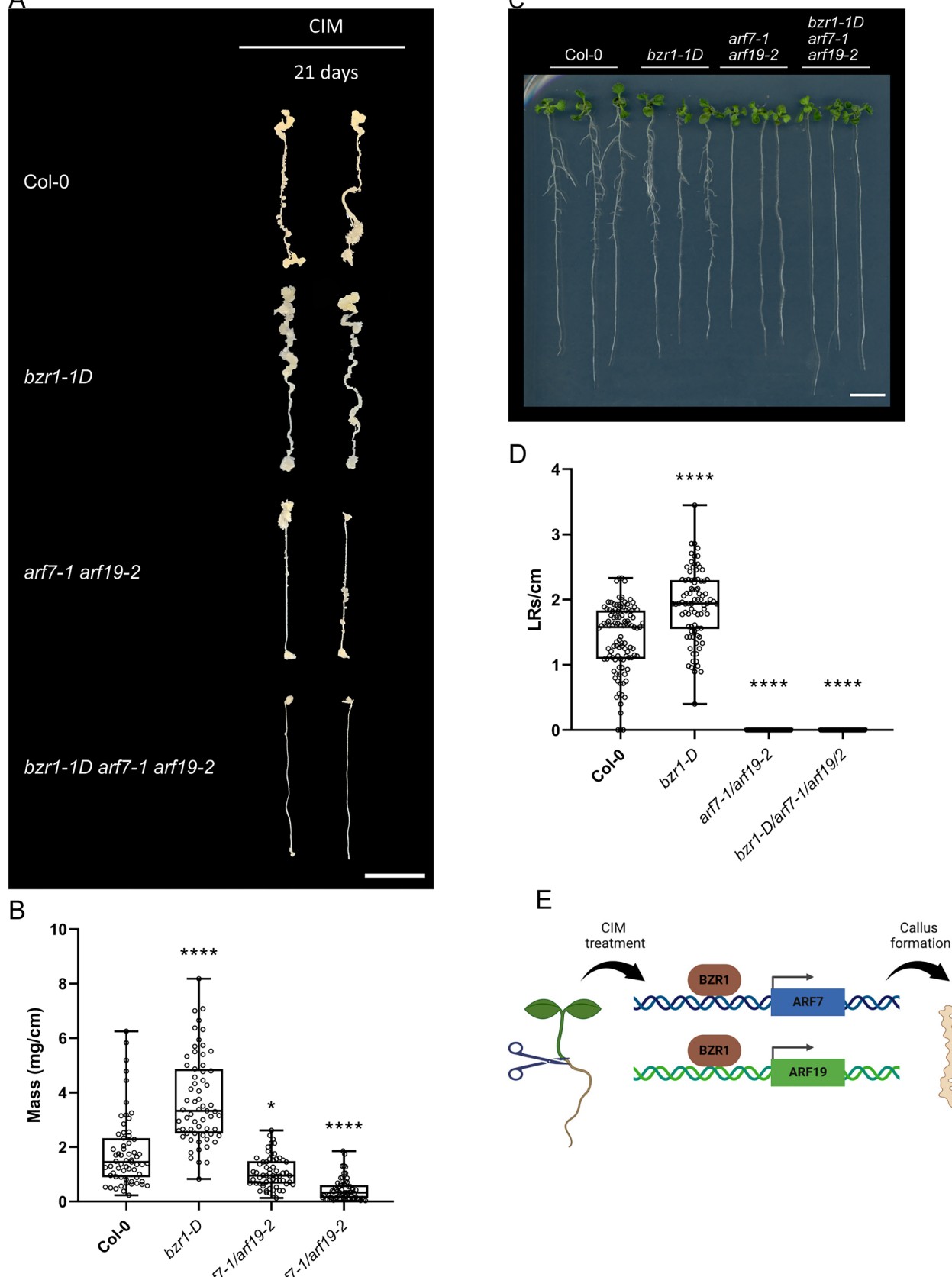

**Figure 6. Enhanced callus and LR formation on bzr1-d are dependent on *ARF7* and ARF19.**

(A) Representative images of callus forming root explants Col-0, bzr1-D, arf7-1/arf19-2 and bzr1-D/arf7-1/arf19-2 from 21-day CIM, scale bar = 1 cm. (B) Fresh weight of 21-day root explants on CIM (Col-0 $n = 63$, bzr1-D $n = 63$, arf7-1/arf19-2 $n = 61$, bzr1-D/arf7-1/arf19-2 $n = 63$). (C) Representative images of 9-day-old seedlings Col-0, bzr1-D, arf7-1/arf19-2 and bzr1-D/arf7-1/arf19-2 on ½ MS media. Scale bar 1 cm. (D) Lateral root count of 9-day-old seedlings, Col-0, bzr1-1D, arf7-1/arf19-2 and bzr1-D/arf7-1/arf19-2 (Col-0 $n = 106$, bzr1-D $n = 86$, arf7-1/arf19-2 $n = 36$, bzr1-D/arf7-1/arf19-2 $n = 48$). (E) Graphical abstract; Upon CIM treatment BZR1 binds and directly regulate *ARF7* and ARF19 to enhance callus formation. The figure was drawn using BioRender (www.biorender.com). Data information: (B, D) Statistical analysis was calculated via Kruskal-Wallis nonparametric test ($P < 0.05$) with a post hoc Uncorrected Dunn's test. Asterisks represent the significant difference from Col-0. (B) Col-0 vs bzr1-D (****$P = 1.977 \times 10^{-7}$), Col-0 vs arf7-1/arf19-2 (*$P = 0.0253$), bzr1-D/arf7-1/arf19-2 (****$P = 5.058 \times 10^{-13}$). (D) Asterisks represent the significant difference from Col-0 (****$P < 0.0001$). Col-0 vs bzr1-D (****$P = 9.8351 \times 10^{-6}$), Col-0 vs arf7-1/arf19-2 (****$P = 7.727 \times 10^{-14}$), bzr1-D/arf7-1/arf19-2 (****$P = 5.0 \times 10^{-324}$). Boxplots: centerlines show the median; the box limits the 25th and 75th percentiles; whiskers extend to the minimum and maximum. Experiments were made in $n = 3$ biological replicates with a total of at least 36 explants per genotype. Asterisks indicate significant differences from Col-0 (*$P < 0.05$, ****$P < 0.0001$). Source data are available online for this figure.

phenotype *bzr1-D arf7-1 arf19-2* was also unable to form LRs, a phenotype reminiscent of *arf7-1 arf19-2*. Regarding the above-ground tissue, *bzr1-D/arf7-1/arf19-2* lines were indistinguishable from *arf7-1/arf19-2* (Fig. EV5). These results altogether indicate that *bzr1-D* enhanced pluripotency arises from direct interaction and regulation of *ARF7* and *ARF19*.

## Discussion

Here we show that the BR signaling transcription factors BZR1 and BES1, despite their high degree of homology, promote different outcomes during pluripotency acquisition, with BZR1 promoting and BES1 preventing callus formation. Our live cell imaging data supports this proposition, showing that BZR1 expression is increased but BES1 expression is repressed by callus-inducing media. As BES1 has been shown to interfere with cell cycle progression and promote cell differentiation (Vilarrasa-Blasi et al, 2014) and given our data, it seems very likely BES1's stabilization is detrimental to pluripotency acquisition or maintenance. Future experiments using cell-type specific reporters or single-cell studies could help uncover if the former or the latter are responsible for *bes1-D* callus phenotype. As a caveat to our experiments, the loss-of-function mutants of BES1 and BZR1 do not display major perturbation of callus formation, which could be very likely explained by the high order of redundancy with BEHs, which we show are also induced during callus formation (Fig. EV2). In contrast to our findings, a *BES1* RNAi KD line exhibited a reduction in shoot organogenesis with BES1 expression increasing during CIM- and, in agreement with our findings, SIM incubation (Wu et al, 2022). These contrasting findings can speculatively be explained by differences in tissue types used as explants: hypocotyls (Wu et al, 2022) versus root tissue (this study). However, the specific RNAi line used in that study was reported to affect *BZR1* and possibly BEH2-3 transcripts (Yin et al, 2002), which complicates pinning down the callus effects exclusively to *BES1* inhibition. Unfortunately, analysis of BR- synthesis and signaling mutants like *det2*, *dwf4* or the *bzr*-sextuplet mutants would be very difficult given their reduced primary root and pleiotropic nature (Chen et al, 2019; Kim et al, 2009). Thus, using such mutants would have complicated the interpretation of any data regarding callus formation. The usage of chemical inhibitors of BR synthesis and signaling allowed us to study the effect of BR synthesis and signaling inhibition specifically during callus formation, avoiding the problem of defective root explants as source tissue. With this, our data clarifies that the stabilization of BZR1 bypasses BR synthesis inhibition and functions additively to BIN2 to promote callus formation. The negative effect of BES1 stabilization on callus formation on the other hand is insensitive to BIN2 inhibition and thus there might be a dependency between the two.

Through ChIP and gene expression analysis, we demonstrate that BZR1 binds and activates the expression of known regulators of stem cells like *ARF7* and *SCR* (Tian et al, 2022), but also of novel targets like *RGF* and *ARF19* (Figs. 5D and EV3). In the case of ARF19, is important to highlight that while ARF19 and *ARF7* are known to work redundantly, research has shown that they are under different transcriptional regulation mechanisms, for instance, *ARF7* expression is decreased upon auxin maxima in the root's oscillation zone (Moreno-Risueno et al, 2010) while ARF19 is known to be induced by auxin (Orosa-Puente et al, 2018). Thus, our finding that BZR1 transcriptionally controls both *ARF*s is an important new regulatory layer during pluripotency acquisition and development (Fig. 6E). Precisely, we demonstrate that *bzr1-D* callus phenotype is completely dependent on *ARF7* and *ARF19* function, indicating that potential effect of BZR1 on other callus-related target genes is upstream of those *ARF*s. In line with this, given that BIN2 and BZR1 both depend on *ARF7&19* to promote callus formation and that BIN2 phosphorylates *ARF7&19* and acts independently of BR to promote callus formation (Cho et al, 2014; Lee and Seo, 2017; Matsuzaki et al, 2010), it is tempting to speculate that the positive effect of both BZR1 and BIN2 arise from direct regulation of these ARFs at transcript and protein levels, respectively. Besides its role in promoting callus formation, we also show for the first time that BZR1 also promotes LR formation. BZR1's effect on LR formation has been previously analyzed and no differences were found regarding Col-0 (Cho et al, 2014). Differences between that report and ours could very well be explained by different growth or experimental conditions but given the similarities between callus and LR formation especially in their reliance on *ARF7* and *ARF19* signaling, it is reasonable to assume that BZR1 promotes both developmental features as we show here. In support of this, previous studies have shown that exogenous application of low BR promote LR formation (Rovere et al, 2022), and importantly, overexpression of *BZR1* in rice also leads to enhanced LR formation (Hou et al, 2022).

In conclusion, we provide compelling evidence of BZR1's function in pluripotency acquisition through direct regulation of *ARF7* and *ARF19*, which adds a novel layer of complexity to the interplay between auxin and BR signaling during development.

# Methods

## Reagents and tools table

| Reagent/resource | Reference or source | Identifier or catalog number |
| --- | --- | --- |
| Experimental models | | |
| Col-0 | | |
| *bzr1-D* | NASC | ABRC stock number: CS65987 |
| *bes1-D* | Ana Caño-Delgado, Crag Genomica | • *bes1-D (Col-0)*: https://doi.org/10.1073/pnas.0906416106 |
| *bes1-ko* | NASC | salk_098634 |
| *bzr1-c1* | This study | |
| *bzr1-c2* | This study | |
| pBES1::BES1-YFP | Jia Li, School of Life Science | https://doi.org/10.1038/s41467-019-12118-4 |
| pBZR1::BZR1-YFP | Wenqiang Tang, Hebei Normal University | https://doi.org/10.1038/s41467-019-12118-4 |
| pUBQ10::YFP-GOT1 | NASC | NASC stock number: N781547 https://doi.org/10.1111/j.1365-313X.2009.03851.x |
| *arf7-1/arf19-2* | NASC | ABRC stock number: CS24630 |
| *bzr1-D/arf7-1/arf19-2* | This study | |
| Recombinant DNA | | |
| Antibodies | | |
| Oligonucleotides and other sequence-based reagents | | |
| Name | | Sequence |
| q_act2_F | | 5′-CTT GTT CCA GCC CTC GTT TGT G-3′ |
| q_act2_R | | 5′-CCT TGG AGA TCC ACA TCT GCT G-3′ |
| q_arf19_F | | 5′-ACA GCT CGA AGA TCC GCT AAC-3′ |
| q_arf19_R | | 5′-TGC ACG CAG TTC ACA AAC TCT TC-3′ |
| q_arf7_F | | 5′-TCA AGG TCA CAG TGA GCA AGT CG-3′ |
| q_arf7_R | | 5′-TGT GGA GCA TGC ATA TGA GCT TGG-3′ |
| q_BEH1_F | | 5′-TTT GTC TTG AAG CTG GTT GGA TCG-3′ |
| q_BEH1_R | | 5′-TTC TGT TGG TCG AGA ACC TTT TC-3′ |
| q_BEH2_F | | 5′-ATG GCA CCA CTT ATC GCA AGG G-3′ |
| q_BEH2_R | | 5′-TGG TAC GAA GGT GCA GGA CTT G-3′ |
| q_BEH3_F | | 5′-TGC AAT GAA GCT GGT TGG ACT G-3′ |
| q_BEH3_R | | 5′-TCC ATT GGT TTG CAT CCC TTG C-3′ |
| q_BEH4_F | | 5′-TGA TGG AAC TAC TTA CCG CAA GGG-3′ |
| q_BEH4_R | | 5′-AGC ACA GGG ACT TGG CTG ATA G-3′ |
| sgRNA2_F | | 5′-ATT GCA GTT GTC CAG TTA CCC CAC-3′ |
| sgRNA2_R | | 5′-AAA CGT GGG GTA ACT GGA CAA CTG-3′ |
| sgRNA3_F | | 5′-ATT GGA GAA AGG GAG AAT AAT CGG-3′ |
| sgRNA3_R | | 5′-AAA CCC GAT TAT TCT CCC TTT CTC-3′ |
| Geno_sgRNA2_F | | 5′-GTT GGA GTA TCA GGG ATG CA-3′ |
| Geno_sgRNA2_R | | 5′-CCA CGA GCC TTC CCA TTT C-3′ |
| Geno_sgRNA3_F | | 5′-GAG AAA AAG AGA GA TTC TTC-3′ |
| Geno_sgRNA3_R | | 5′-TGA AAG AGG GCT CTG GTT CT-3′ |

| Reagent/resource | Reference or source | Identifier or catalog number |
| --- | --- | --- |
| Geno_arf7Salk_F | | 5′-CAG CTA GAT CGT TCG AAA TGG-3′ |
| Geno_arf7Salk_R | | 5′-AGC ACA TCA CCA TTT AGG TGC-3′ |
| Geno_arf19Salk_F | | 5′-TTG GAG TTG CTG AGG ATT TTG-3′ |
| Geno_arf19Salk_R | | 5′-TTT GAG ACT GAG GAT TGT GGG-3′ |
| LBb1.3_F | | 5′-ATT TTG CCG ATT TCG GAA C-3′ |
| Geno_bzr1-Ddecapping_F | | 5′-CAC ATC GCC ACC AGT TTC ATA CCA-3′ |
| Geno_bzr1-Ddecapping_R | | 5′-TAT CCT CTC TCC TTC CCA-3′ |
| p_ARF7 P1_F | | 5′-GAG CTT CGT TAA AAA CGG AAT C-3′ |
| p_ARF7 P1_R | | 5′-CTC AAA GAG ATC AAA AAC CAC G-3′ |
| p_ARF7 P2_F | | 5′-CAA GGT CAC AGT GAG CAA G-3′ |
| p_ARF7 P2_R | | 5′-TGG AAG GAA GAT TCG GGT AAC-3′ |
| p_ARF19 P1_F | | 5′-ACG ATG AGG AGA AGA GAC ATA GAG G-3′ |
| p_ARF19 P1_R | | 5′-ACG TGA GGT GTG GCA GAG G-3′ |
| p_ARF19 P2_F | | 5′-AGA GAG CTG AAA GTA AGG AAC AAC TC-3′ |
| p_ARF19 P2_R | | 5′-CAG AGC TAC GGA CAT CAT TAT ACT CC-3′ |
| p_RGF1 P1_F | | 5′-CCC AAG ATT CAA ATG TCA TGC TCC G-3′ |
| p_RGF1 P1_R | | 5′-AGA AAA GGG AAG CAA GTG TGC AG-3′ |
| p_RGF1 P2_F | | 5′-AGT ATC CGA AAC GAC TGC G-3′ |
| p_RGF1 P2_R | | 5′-AAA GAA AAT TAA ACC AGC ACC GGC-3′ |
| p_SCR_F | | 5′-TGG ACT TGG AGA AAG ACA TTC AGC-3′ |
| p_SCR_R | | 5′-TGA ACC CAA GAA GAA ACC ATC CAC-3′ |
| qCHIP-AGL91AT3G66656_F | | 5′-GTT CAC AAG GAC AGT ATT CTC-3′ |
| qCHIP-AGL91AT3G66656_R | | 5′-ACA TCA ACA TCA ACA TCA TCA-3′ |
| Chemicals, enzymes, and other reagents | | |
| NcoI | Thermo Fischer | ER0571 |
| 2,4-D | Thermo Fischer | A12467.18 |
| Kinetin | Sigma-Aldrich | K3378 |
| BAP | Sigma-Aldrich | B3408 |
| IAA | Sigma-Aldrich | I2886 |
| Bikinin | Sigma-Aldrich | SML0094 |
| Brassinazole | Sigma-Aldrich | SML1406 |
| TRIzol™ Reagent | Invitrogen | 15596026 |
| DNase I | Thermo Fischer | EN0525 |
| RevertAid First Strand cDNA Synthesis Kit | Thermo Fischer | K1622 |
| Fast SYBR™ Green Master Mix | Thermo Fischer | 4385612 |
| GFP beads | ChromoTek | https://www.ptglab.com/products/GFP-Trap-Agarose-gta.htm?srsltid=AfmBOopF_voPCJSLaPMr2BqHg-JcpTxw7E63dyFdBRIOq9jFFrfqanbN |
| T4 DNA Ligase | Thermo Fischer | EL0011 |
| Software | | |
| ImageJ | | |
| Quantstudio 1 | | |
| Quantstudio 5 | | |

| Reagent/resource | Reference or source | Identifier or catalog number |
|---|---|---|
| CRISPR-P 2.0 | http://crispr.hzau.edu.cn/CRISPR2/ | |
| ZEN2012 (Zeiss) | | |
| Graphpad Prism 8 | | |
| Other | | |

## Plant growth and phenotypic characterization

For growth of Arabidopsis seedlings, ½ MS media (0.8% agar, 1% sucrose at 5.7 pH) was used, exposing the plants to a photoperiod of 16 h (120 μE/m²/s), at 21 C°). Seed sterilization was accomplished with 1.3% bleach for 2 min, followed by 70% ethanol for 1 min, and washed three times with sterile water. We used following lines in this study: Col-0, *bzr1-D* (Wang et al, 2002), *bes1-D* (Ibañes et al, 2009), *arf7-1 arf19-2*(NASC), *bzr1-D arf7-1 arf19-2* (crossed), *pBZR1::gBZR1-YFP* (Chen et al, 2019), *pBES1::gBES1-YFP* (Chen et al, 2019), *pUBQ10::YFP-GOT1* (YFP fusion to GOT1 (At3g03180), a gene involved in vesicle trafficking between Golgi and ER) (Geldner et al, 2009), *bzr1-c1* (constructed), *bzr1-c2*(constructed) and *bes1-ko* (salk_098634). The *bzr1-D/arf7-1/arf19-2* crosses were made and genotyped using Salk primer LbB1.3 and gene-specific primers for *arf7-1* and *arf19-2*, and decapping primers for *bzr1-D*. After digesting with the NcoI (Thermo Scientific) restriction enzyme, a 25 bp band shift was observed in the *bzr1-D* mutant.

For experiments, 7-day-old seedlings were grown on solid half-strength MS before the root was excised. For callus growth, explants were put on solid CIM media (3,1 g/L Gamborg's B5 salts with vitamins, 20 g/L Glucose, 0.5 g/L MES, 8 g/L Agar, pH adjusted to 5.7 with 1 M KOH) containing 2,4-D (0.5 mg/L) and kinetin (0.05 mg/L). Incubation time was 21 days before moving them to solid SIM media (4.3 g/L Murashige and Skoog, 30 g/L sucrose, 8 g/L Agar, pH adjusted to 5,8 with 1 M NaOH) containing BAP (1.0 mg/L) and IAA (0.1 mg/L). Calli were imaged and had their mass quantified after 21 days on CIM and after 21 days on SIM.

For the inhibitor assay, 5 μM Bikinin and 2 μM Brassinzole (Sigma-Aldrich) were used. Both were solubilized in DMSO and added to the CIM media, including a DMSO mock treatment, as described above.

LR scoring was performed as described by (Ebstrup et al, 2024). In brief, seedlings were germinated on ½ MS media and set to grow vertically. LR's were counted at day 9 as described by (Dubrovsky and Forde, 2012), and only LR's that had at least reached stage IV were included.

## Microscopy and image analysis

Confocal Images of root explants were taken post 6-day treatment with either liquid half-strength MS or liquid CIM media. For measurements on SIM, explants were grown 21 days on CIM and then transferred to SIM media for 3 days. Quantification of mean gray value of pBES1::BES1-YFP and pBZR1::BZR1-YFP in the midlayer (indicated by white arrows) and the root tip with signal were measured using ImageJ software.

Confocal microscopy images were taken with LSM700 Zeiss confocal microscopy using ×20 and ×63 objectives for Fig. 4A,C and 4B,E,F, respectively. Images were processed using ZEN2012 (Zeiss) and intensity were quantified using mean gray value in ImageJ software.

## RNA extraction and qPCR

Total RNA was extracted from samples as previously described in (Zuo et al, 2022) through TRIzol reagent (Thermo Scientific), treated with DNase (Thermo Scientific) and cDNA was reverse transcribed through RevertAid kit (Thermo Scientific) following the manufacturer descriptions. For rt-qPCR Fast Sybr green mastermix (Thermo Fisher) was used and the reaction was performed on Quantstudio 1 and Quantstudio 5 (Applied Biosystems). Actin2 (AT3G18780) was used as an internal control. Primers used can be found in the Reagents and Tools Table.

## Chromatin immunoprecipitation (ChIP)

The ChIP was carried out using the Chromotek Chromatin Immuno-precipitation protocol for *A. thaliana* (Gendrel et al, 2005; Schuster et al, 2015). Approximately 2 g of fresh-weight material per genotype was used, and samples were crosslinked in vivo for 15 min with formaldehyde (1%). After sonication (500–1000 pb fragments), DNA concentration was measured, and all the samples were diluted to the same DNA concentration. 20 μL (ChromoTek) GFP beads were added, and samples were incubated overnight at 4 °C with gently agitation. De-crosslinked samples were then purified and used for real-time quantitative PCR. ChIP-RT-PCR data was generated from at least two independent experiments. Primers are listed in the Reagents and Tools Table. Enrichment was calculated relative to the input and expressed as fold change to the pUBQ10::YFP-GOT1 which was arbitrarily set to 1. For ChIP-seq, preparation of samples was done as described above with 21 d callus explants from BZR1-YFP and YFP-GOT1 and Sequencing of input and immunoprecipitated genomic DNA, as well as raw data processing (including trimming with Skewer, quality control with FastQC, mapping with BWA), were carried out by Novogene.

## Crispr–Cas9 construction

SgRNA was designed with the help of http://crispr.hzau.edu.cn/CRISPR2/ targeting the 5' end of AT1G75080. Two primer sets were used. The primer pair was annealed, followed by ligation using T4 ligase (Thermo Scientific) into pHEE401(pre-digested with bsa1) before transformation into *E. coli*. Sequencing was done before transformation sgRNA into GV3101. Construct was moved to Agrobacterium followed by transformation into plants by flower dipping method. Genotyping primers and sgRNA primers are listed in the Reagents and Tools Table.

## Statistical analysis

Data was obtained and analyzed without blinding. Removal of data points was only performed in cases obvious genotype contamination. Data were subjected for normality testing using Shapiro–Wilks and Kolmogorov–Smirnov tests and appropriate

testing was selected accordingly. All statistical tests were performed using GraphPad Prism 8.

## Data availability

The source data for the microscopy has been deposited at BioImage Archive with accession number: S-BIAD1610. https://www.ebi.ac.uk/biostudies/BioImages/studies/S-BIAD1610.

The source data of this paper are collected in the following database record: biostudies:S-SCDT-10_1038-S44319-025-00433-5.

## Peer review information

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

## Acknowledgements

The authors thank Ana Caño-Delgado, Crag Genomica, for the *bes1-D* seeds, Wenqiang Tang, Hebei Normal University, for the *pBZR1::gBZR1-YFP* seeds, and Jia Li, School of Life Science for the *pBES1::gBES1-YFP* seeds. Confocal microscopy imaging was performed using equipment from the Center for Advanced Bioimaging (CAB) Denmark, University of Copenhagen. This work was funded by the Danish Research Agency grant to ER (DFF1- 1032-00249B) and the Novo Nordisk Fonden (NNF190C0055222) to MP.

## Author contributions

**Thomas Ammitsøe**: Data curation; Formal analysis; Validation; Investigation; Visualization; Methodology; Writing—original draft; Writing—review and editing. **Elise Ebstrup**: Data curation; Formal analysis; Validation; Investigation; Visualization; Methodology; Writing—original draft; Writing—review and editing. **Noel Blanco-Touriñán**: Validation; Investigation; Visualization; Methodology; Writing—original draft; Writing—review and editing. **Julie Hansen**: Validation; Investigation; Visualization; Methodology; Writing—original draft; Writing—review and editing. **Christian S Hardtke**: Resources; Supervision; Validation; Writing—original draft; Writing—review and editing. **Morten Petersen**: Conceptualization; Supervision; Funding acquisition; Visualization; Writing—original draft; Project administration; Writing—review and editing. **Eleazar Rodriguez**: Conceptualization; Data curation; Formal analysis; Supervision; Funding acquisition; Validation; Visualization; Writing—original draft; Project administration; Writing—review and editing.

Source data underlying figure panels in this paper may have individual authorship assigned. Where available, figure panel/source data authorship is listed in the following database record: biostudies:S-SCDT-10_1038-S44319-025-00433-5.

## Disclosure and competing interests statement

The authors declare no competing interests.

# Expanded View Figures

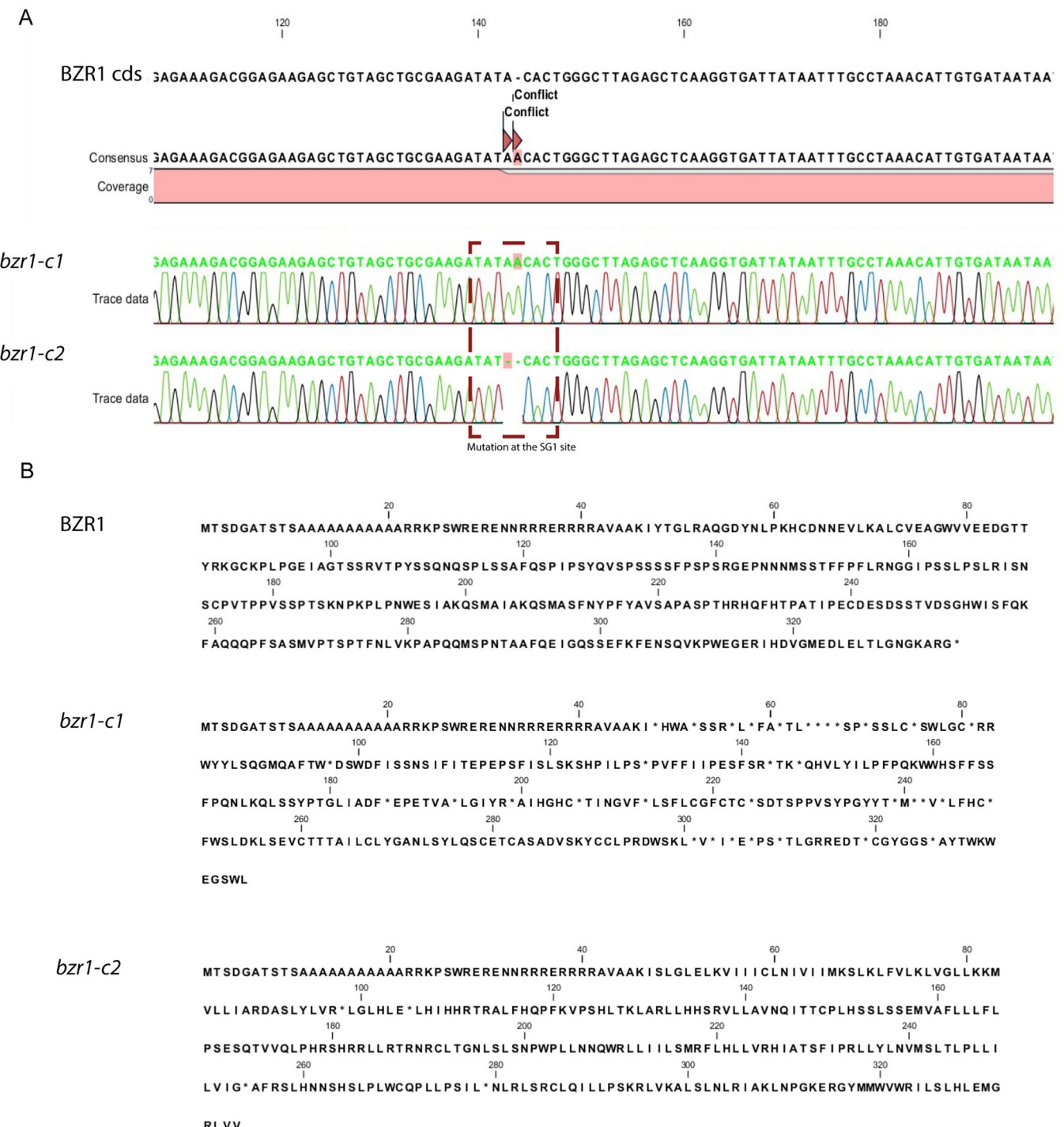

**Figure EV1. Generation of *bzr1* CRISPR lines.**

(A) *bzr1-c1* and *bzr1-c2* lines. DNA sequence for *BZR1* CDS and the two CRISPR–Cas9 generated BZR1 mutant lines *bzr1-c1* and *bzr1-c2* showing mutations to occur at position 144 and 143 (indicated by dashed square). (B) Protein sequence of translated BZR1 CDS. Frameshift mutation and premature stop codon are observed for *bzr1-c1* and *bzr1-c2*, respectively. See material and method for details.

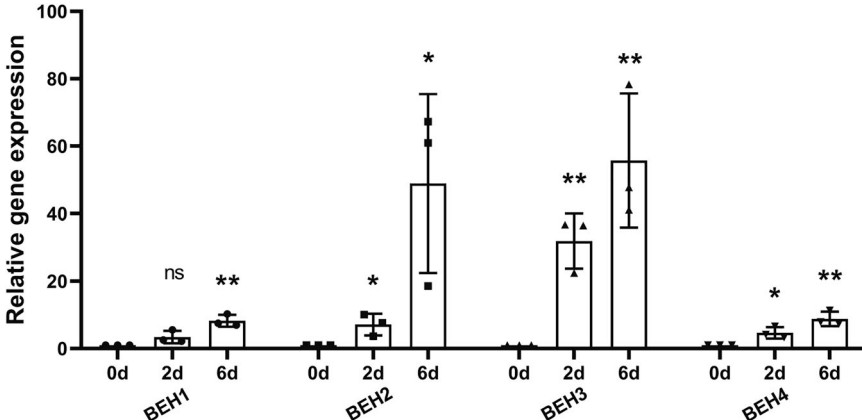

**Figure EV2.   BEH1-4 is transcriptionally induced upon CIM treatment.**

Rt-qPCR testing expression of *BEH1-4* after 2- and 6- days of CIM incubation, compared to 0 d treatment i.e. root explants. Tissue was collected from 7-day old seedlings and put in liquid CIM media for 2 or 6 days before collecting. Expression was normalized to *ACTIN2*. Bars display the mean and SD of $n = 3$ biological replicates each performed with 3 technical replicates. Statistical tests were made against 0 d. Statistical analysis was performed using unpaired Student's *t* test (*$P < 0.05$, **$P < 0.01$). *BEH1*: 0 d vs 2 d (ns $P = 0.087$), 0 d vs 6 d (**$P = 0.0020$). *BEH2*: 0 d vs 2 d (*$P = 0.0304$), 0 d vs 6 d (*$P = 0.0350$). *BEH3*: 0 d vs 2 d (**$P = 0.0030$), 0 d vs 6 d (**$P = 0.0087$). *BEH4*: 0 d vs 2 d (*$P = 0.0196$), 0 d vs 6 d (**$P = 0.0032$). Source data are available online for this figure.

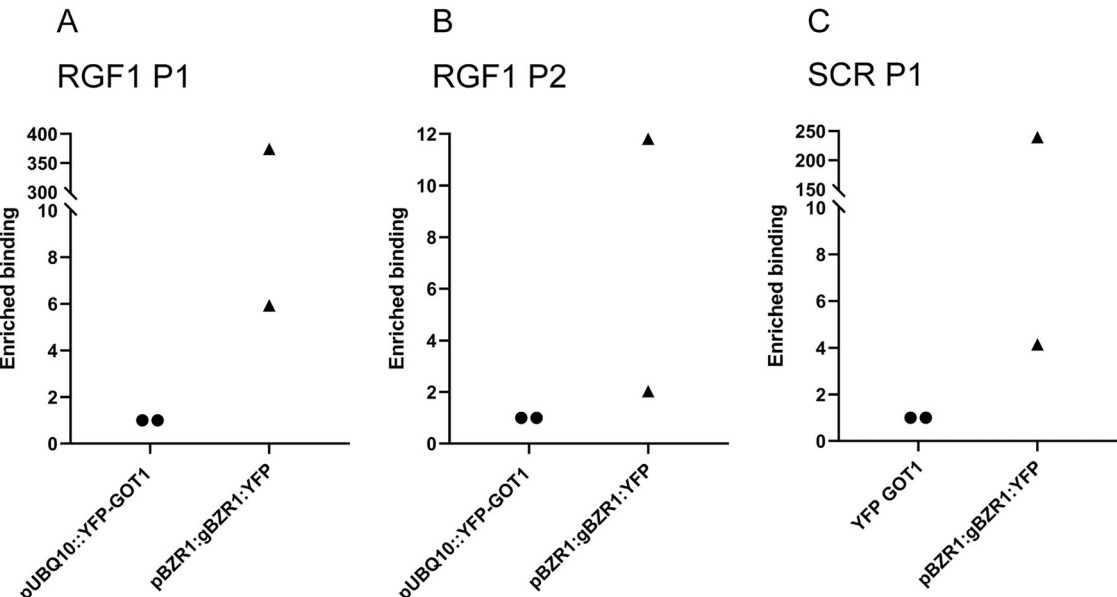

**Figure EV3. BZR1-YFP bind to the promoter of *RGF1* and *SCR1*.**

ChIP experiment showing the DNA binding of BZR1-YFP to RGF1 P1/P2 and *SCR1* promoter region. BZR1-YFP and YFP-GOT1 of callus explants after 21 days on CIM of *RGF1* P1 (**A**), *RGF1 P2* (**B**), *SCR1* (**C**). This was determined through quantitative RT-PCR. Fold change was calculated by normalizing to the YFP-GOT1 control for each sample. For each promoter region $n = 2$ biological replicates are shown. Source data are available online for this figure.

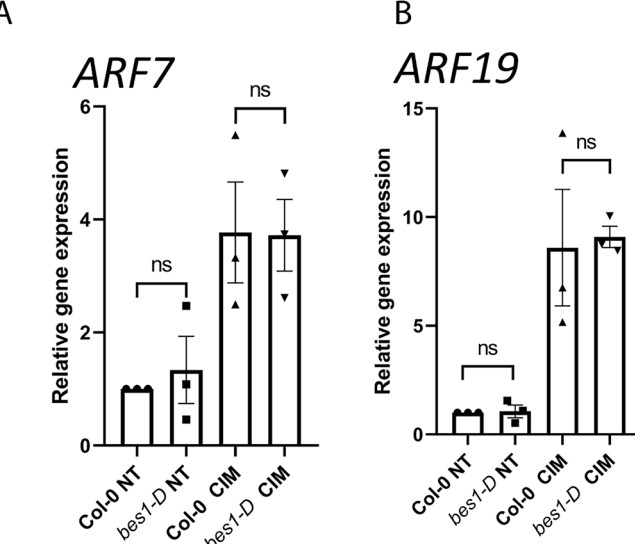

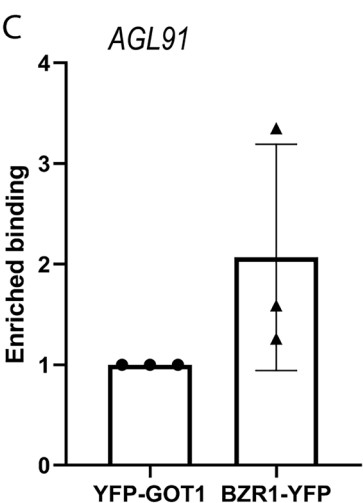

**Figure EV4. Expression of *ARF7* and *ARF19* in *bes1-D* callus and ChIP-RT-PCR negative control.**

(A, B) Relative expression of A) *ARF7* and B) *ARF19* were analyzed in Col-0 and *bes1-D*. Tissue was obtained from NT and 21-day CIM-treated root explants. Expression was normalized to *ACTIN2* and is relative to Col-0 NT; i.e. root explants from 7-day old seedlings grown on MS media. Bars display the mean and SEM of $n = 3$ biological replicates each performed with 3 technical replicates. Significance was calculated via student *t* test. Asterisks represent the significant difference from Col-0 NT/CIM from each treatment (*$P < 0.05$). (A) *bes1-D* NT vs Col-0 NT (ns $P = 0.60$), *bes1-D* CIM vs Col-0 CIM (ns $P = 0.9646$). (B) *bes1-D* NT vs Col-0 NT (ns $P = 0.8537$), *bes1-D* CIM vs Col-0 CIM (ns $P = 0.8644$). (C) Negative control of BZR1 ChIP-The DNA binding of BZR1-YFP to *AGL91* promoter region was determined through quantitative RT-PCR. Bars display the mean and SD of $n = 3$ biological replicates each performed with 3 technical replicates. Significance was calculated via Student's *t* test. BZR1-YFP vs YFP-GOT1 (ns $P = 0.175$), with ns representing a non-significant difference from YFP-GOT1 (control). Source data are available online for this figure.

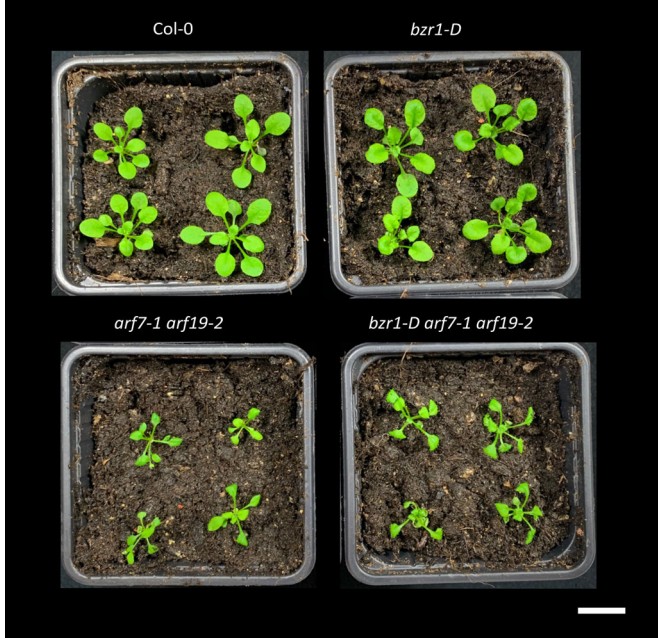

**Figure EV5. Aboveground tissue of *bzr1-D/arf7-1/arf19-2* displays a similar phenotype as *arf7-1/arf19-2*.**

Representative pictures of Col-0, *bzr1-D*, *arf7-1/arf19-2*, *bzr1-D/arf7-1/arf19-2* phenotypes of 2-week-old plants grown under standard conditions (See materials and methods). The observed phenotypes were consistent across multiple pots.

