## [Peer Review File · EMBO Reports]

BZR1 promotes pluripotency and callus formation through direct regulation of ARF7 & ARF19

Thomas Ammitsøe, Elise Ebstrup, Noel Blanco-Tourinan, Julie Hansen, Christian Hardtke, Morten Petersen, and Eleazar Rodriguez

Corresponding author(s): Eleazar Rodriguez (eleazar.rodriguez@bio.ku.dk), Morten Petersen (shutko@bio.ku.dk)

Review Timeline:

Transfer Date:	12th Nov 24
Editorial Decision:	13th Dec 24
Revision Received:	7th Feb 25
Editorial Decision:	18th Feb 25
Revision Received:	7th Mar 25
Accepted:	12th Mar 25

Editor: Achim Breiling

Transaction Report: This manuscript was transferred to EMBO reports following peer review at Review Commons.

The logo for Review Commons, featuring the word "Review" in a large, blue, serif font with a diagonal slash through the letter 'V', and the word "COMMONS" in a smaller, blue, sans-serif font below it.

Review #1

1. Evidence, reproducibility and clarity:

Evidence, reproducibility and clarity (Required)

In this manuscript, the authors explore the novel roles of BZR1 and BES1 in callus formation. They demonstrate that the gain-of-function mutant *bzr1-1D* shows increased callus formation, whereas the *bes1-D* mutant displays decreased callus formation, attributable to differential expression patterns of BZR1 and BES1. Additionally, the authors illustrate that BZR1 directly binds to the promoters of ARF7 and ARF19, inducing their expression and consequently promoting callus formation. While the authors provide genetic and molecular evidence to support their assertions, there are significant concerns that must be addressed prior to publication.

1. The authors demonstrate that BZR1 and BES1 play opposing roles in plant callus formation. Is this regulation reliant on BR for its function, or do BZR1/BES1 serve specific functions during callus formation? Is BR implicated in the regulation of callus formation, and do mutants defective in BR synthesis and BR signal transduction, such as *det2*, *dwf4*, *bri1* and *bin2*, exhibit callus formation phenotypes?
2. BZR1 promotes callus formation, while BES1 suppresses it, which is due to their differing expression profiles. What outcomes would arise from swapping promoters, employing the BZR1 promoter to activate *bes1-D* or using the BES1 promoter to drive *bzr1-1D*? Additionally, considering the existence of BZR sextuplet mutants, it is recommended that the authors utilize these variants to delve into the function of BZR family proteins in callus formation.
3. In the manuscript, the authors mentioned that "BZR1 signal appeared to increase upon CIM treatment, while the BES1 signal is reduced. The lower expression of BES1 could be explained by the fact that callus induction is not commonly associated with BR synthesis, as BES1 has been shown to positively regulate this process (Yu et al., 2011) while BZR1 acts on more downstream responsive genes". However, there is no evidence supporting the claim that BES1 induces BR biosynthesis while BZR1 reduces it. Published data suggests that both BZR1 and BES1 negatively regulate the expression of BR biosynthesis genes. Additionally, auxin has been shown to induce the nuclear localization of BZR1 by regulating the activity of 14-3-3 proteins. It has also been reported that 14-3-3 proteins interact with BES1 to promote its cytoplasmic localization. Hence, it is conceivable that auxin also regulates the nuclear localization of BES1.
PS. It should add the page number and line number in the manuscript.
4. The authors note that a previous study demonstrated increased callus formation

phenotypes for both bin2-1 mutants, and in this study, the authors observed a similar increase in callus formation for bzs1-1D mutants. It is suggested that the authors perform additional experiments to evaluate the phenotypes of bin2-1, bzs1-1D, and their double mutants under standardized conditions. This investigation would provide clarity on whether BZR1 and BIN2 operate within the same signal transduction pathway.

5. What are the expression levels of ARF7 and ARF19 in bes1-D mutants? Do the results indicate that BES1 regulates the expression of these two genes? Furthermore, is there evidence supporting the notion that BES1 modulates callus formation by regulating the expression of ARF7 and ARF19?

2. Significance:

Significance (Required)

This draft paper investigates the influence of BZR1 and BES1, essential components in brassinosteroid signal transduction, on plant callus formation. It holds significance in unraveling the molecular mechanisms driving the development of plant cell pluripotency. It will be of interest to audiences engaged in the realms of plant hormones and pluripotency.

3. How much time do you estimate the authors will need to complete the suggested revisions:

Estimated time to Complete Revisions (Required)

(Decision Recommendation)

Between 3 and 6 months

Yes

Review #2

1. Evidence, reproducibility and clarity:

Evidence, reproducibility and clarity (Required)

This manuscript entitled "BZR1 promotes pluripotency acquisition and callus development through direct regulation of ARF7 and ARF19" revealed that gain-of-function mutants *bzr1-D* and *bes1-D* exhibit altered callus formation, yet disruption of these transcription factors does not produce major changes to callus formation or de novo organogenesis. The authors further studied the regulatory mechanism that BZR1 directly binds to the promoters of ARF7 and ARF19, which enhances their transcription and influences callus formation and LR development. Generally, the topic is interesting, but the data is some kind of simple and preliminary. More evidences are required and several issues needs to be addressed before being considered for publishing. The main concerns (not all) are listed below.

1. BZR1s are the most important transcription factors transcriptional regulating BR-responsive genes expression. So, it is only BZR1 and BES1 involving in pluripotency acquisition and callus development? How about other BR mutant including deficient (*det2*, *cpd*, *dwf4*?) and insensitive mutants (*bri1*, *bin2*, *bsk*?)? Is it BR or BZR1s involving in pluripotency acquisition and callus development?
2. The author mentioned in both abstract and results (Figure 1) that *bzr1-D* and *bes1-D* exhibit different phenotypes, which is very interesting. Both *bzr1-D* and *bes1-D* can cause an increased BR signal in plants, but their phenotypes on CIM are completely different. The authors try to explain that BZR1 expresses in callus but BES1 not. Does the expression difference between these two genes determine their different function in callus development? Besides, the expression pattern reflected by marker lines in Figure don't illustrate significant differences (semi-quantitative statistical analysis is needed). The staining of cell wall is helpful to see the difference between BES1 and BZR1. Furthermore, the developmental stages are not the same in BES1 and BZR1 lines in Figure 3C.
3. Dose *bzr1-1D* and *bes1-1D* has different BR level or auxin level? After CIM, did BR or auxin level change in Col, *bzr1-1D* and *bes1-D*?
4. The authors illustrated that *bzr1c* and *bes1-ko* has not phenotypes. BZR family has 6 member and there are published mutant of bzrs. It is necessary to check its phenotype to gain clear conclusion.
5. It is not surprising that BZR1 can directly bind to ARF7/19, as previously reported. Have the authors detected genes related to other callus attributes, such as WOX5, PLT, SCR, etc? The authors have BZR1 tag line, which could be used to do CHIP-seq of BZR1 to identified more downstream targets. Combining with RNA-seq, the big picture and detailed mechanism could be discovered.
6. The authors organize the regulation of lateral organ development and callus

development. It is not complete same mechanism between these two things.

7. What is the mechanism of increased bzm1-1D and bsm1-D shoot in SIM (Figure 2D)? What are the downstream target genes that they regulate shoot growth?

2. Significance:

Significance (Required)

The current version of this manuscript is preliminary and needs substantive improvement.

3. How much time do you estimate the authors will need to complete the suggested revisions:

Estimated time to Complete Revisions (Required)

(Decision Recommendation)

Between 3 and 6 months

Yes

Rebuttal

Reviewer #1 (Evidence, reproducibility and clarity (Required)):

In this manuscript, the authors explore the novel roles of BZR1 and BES1 in callus
formation. They demonstrate that the gain-of-function mutant *bzr1-1D* shows increased
callus formation, whereas the *bes1-D* mutant displays decreased callus formation,
attributable to differential expression patterns of BZR1 and BES1. Additionally, the authors
illustrate that BZR1 directly binds to the promoters of ARF7 and ARF19, inducing their
expression and consequently promoting callus formation. While the authors provide genetic
and molecular evidence to support their assertions, there are significant concerns that must
be addressed prior to publication.

**R: We thank the reviewer for their positive appreciation of our story.**

1. The authors demonstrate that BZR1 and BES1 play opposing roles in plant callus
formation. Is this regulation reliant on BR for its function, or do BZR1/BES1 serve specific
functions during callus formation? Is BR implicated in the regulation of callus formation, and
do mutants defective in BR synthesis and BR signal transduction, such as *det2*, *dwf4*, *bri1*
and *bin2*, exhibit callus formation phenotypes?

**R: We agree with the reviewer that these are interesting questions to answer.**
**Accordingly, we have performed several experiments to address this issue. As**
**explained in the manuscript, such mutants (L198-200; 365-368) would be problematic**
**to analyze due to reduced primary root length and pleiotropisms which would have**
**complicated data interpretation. Alternatively, we decided to use chemical treatments**
**to interfere with BR signalling (Bikinin-Bik) and biosynthesis (Brassinazole- BRZ),**
**specifically during callus formation. Our results show that Bik treatment led to**
**decreased callus mass in both Col-0 and *bzr1-D*, indicating that GSK3 kinases**
**positively impact callus formation. This is consistent with the data from (Lee and Seo,**
***Planta* 246, 797–802, 2017) showing that *bin2-1* gain of function produced more callus.**
**Importantly, Bik treated *bzr1-D* explants produced significantly higher callus mass**
**than Bik-treated Col-0, being at a similar level to untreated Col-0 callus masses. This**
**suggests that the positive effects of GSK3 kinases like BIN2 and BZR1 on callus**
**formation are additive.**

**Regarding BRZ treatment, we observed a significant decrease in callus formation in**
**Col-0 explants, but the treatment did not affect *bzr1-D*. These results indicate that BR**
**synthesis contributes to callus formation and that BR synthesis inhibition can be**
**bypassed by gain of function of BZR1.**

**As for *bes1-D*, neither of the treatments caused significant changes in callus masses.**
**Taken together, these data indicates that while *bzr1-D* remains partially sensitive to**
**changes in BIN2's activity or signaling, likely in an indirect manner, *bes1-D* is largely**
**autonomous. This might not be surprising given that BES1 has been shown to be**
**dephosphorylated and activated in a BR-independent manner (Albertos et al 2022,**
**EMBOJ). We have added this to the discussion (L222 and 379)**

2. BZR1 promotes callus formation, while BES1 suppresses it, which is due to their
differing expression profiles. What outcomes would arise from swapping promoters,
employing the BZR1 promoter to activate *bes1-D* or using the BES1 promoter to drive *bzr1-*
*1D*? Additionally, considering the existence of BZR sextuplet mutants, it is recommended
that the authors utilize these variants to delve into the function of BZR family proteins in

callus formation.

**R: These are an interesting set of questions which we will break down as follows:**

**Concerning promoter swapping, we understand the rationale behind this comment**
**and its potential interest. However, we notice that we detect an enrichment for ARF7**
**and ARF19 promoters binding by BZR1 but not BES1 (Figure 5 and Figure R1, this**
**letter). This strongly suggests a different molecular basis for diversification of**
**regulatory function of BZR1 and BES1. Therefore, swapping promoters, while**
**interesting, would probably not provide significant new clarifications in this context,**
**besides being a time-consuming endeavor that would substantially delay publication.**

**Regarding the bzt sextuplet mutant, we agree that it would be interesting to verify the**
**extent of redundancy within the BZR/BES/BEH family. In that line, our novel qRT-PCR**
**data (Figure S2) shows that all BEHs are induced upon treatment with CIM and given**
**our callus data, it seems likely that the whole BZR1 family might play a role in callus**
**formation. Unfortunately, the sextuplet mutant is very pleiotropic, displaying several**
**BR independent phenotypes, is severely dwarfed, sterile and has a miniscule primary**
**root (Chen et al., 2019 Molecular Plant). Obtaining seeds from this mutant in sufficient**
**amount to generate calli from such small roots would be extremely laborious and time**
**consuming. That data might even confound interpretation on the function of BEHs**
**role due to the pleiotropic effects and produce only minor gains to our understanding**
**of the role of BR signaling in callus formation. We trust that the new data generated**
**with chemical inhibition of BR synthesis and Bixinin (Figure 3) sufficiently clarifies**
**how BZR1 functions within a BR context of callus formation.**

Figure for referee with unpublished data and its description has been removed upon request by the authors.

3. In the manuscript, the authors mentioned that "BZR1 signal appeared to increase upon
CIM treatment, while the BES1 signal is reduced. The lower expression of BES1 could be
explained by the fact that callus induction is not commonly associated with BR synthesis, as
BES1 has been shown to positively regulate this process (Yu et al., 2011) while BZR1 acts
on more downstream responsive genes". However, there is no evidence supporting the
claim that BES1 induces BR biosynthesis while BZR1 reduces it. Published data suggests
that both BZR1 and BES1 negatively regulate the expression of BR biosynthesis genes.
Additionally, auxin has been shown to induce the nuclear localization of BZR1 by regulating
the activity of 14-3-3 proteins. It has also been reported that 14-3-3 proteins interact with
BES1 to promote its cytoplasmic localization. Hence, it is conceivable that auxin also
regulates the nuclear localization of BES1.

PS. It should add the page number and line number in the manuscript.

**R: We thank the reviewer for this comment, we have modified the text accordingly as**
**our data does indicate that high auxin levels on CIM interfere with BES1 expression**
**(L235).**

4. The authors note that a previous study demonstrated increased callus formation
phenotypes for both *bin2-1* mutants, and in this study, the authors observed a similar
increase in callus formation for *bzr1-1D* mutants. It is suggested that the authors perform
additional experiments to evaluate the phenotypes of *bin2-1*, *bzr1-1D*, and their double
mutants under standardized conditions. This investigation would provide clarity on whether
BZR1 and BIN2 operate within the same signal transduction pathway.

**R: We acknowledge this is a valid suggestion. Due to sterility of *bin2-1* and the time-**
**consuming nature of performing crosses and generating tissue to perform this**
**analysis, we decided to use the GSK3 inhibitor Bik. Please refer to our answer above**
**regarding those results and the inferred synergy of BIN2 and BZR1 signalling to**
**induce callus formation and its potential BR-independency.**

5. What are the expression levels of ARF7 and ARF19 in *bes1-D* mutants? Do the results
indicate that BES1 regulates the expression of these two genes? Furthermore, is there
evidence supporting the notion that BES1 modulates callus formation by regulating the
expression of ARF7 and ARF19?

**R: We thank the reviewer for this interesting question. From our CHIP-rtQPCR data,**
**we do not find significant enrichment of ARF7 or ARF19 DNA in BES1 pulldowns,**
**which is contrary to what we see with BZR (Figure 1). We also tested the expression**
**of ARF7/19, using rt-qPCR in the *bes1-D* line after CIM treatment and observed no**
**difference to Col-0, indicating that *bes1-D* does not regulate *ARF7/19* at the**
**transcriptional level (Figure S3 A-B). Moreover, BES1 expression is reduced in CIM**
**(Figure 4) and *bes1-D* has reduced callus mass (Figure 1). Collectively, our data**
**suggests that BES1 does not regulate callus formation via ARF7 and ARF19.**
**Moreover, and as we state in the discussion (L358), BES1 is known to promote**

**differentiation of stem cells by direct regulation of BRAVO (Vilarrasa-Blasi 2014 Dev.**
**Cell), being that this is a more likely explanation of the reduced callus mass seen in**
**bes1-D.**

Reviewer #1 (Significance (Required)):

This draft paper investigates the influence of BZR1 and BES1, essential components in
brassinosteroid signal transduction, on plant callus formation. It holds significance in
unraveling the molecular mechanisms driving the development of plant cell pluripotency. It
will be of interest to audiences engaged in the realms of plant hormones and pluripotency.

**R: We thank the reviewer for such a positive evaluation of our findings and the**
**relevance to the field.**

Reviewer #2 (Evidence, reproducibility and clarity (Required)):

This manuscript entitled "BZR1 promotes pluripotency acquisition and callus development
through direct regulation of ARF7 and ARF19" revealed that gain-of-function mutants bsr1-D
and bes1-D exhibit altered callus formation, yet disruption of these transcription factors does
not produce major changes to callus formation or de novo organogenesis. The authors
further studied the regulatory mechanism that BZR1 directly binds to the promoters of ARF7
and ARF19, which enhances their transcription and influences callus formation and LR
development. Generally, the topic is interesting, but the data is some kind of simple and
preliminary. More evidences are required and several issues needs to be addressed before
being considered for publishing. The main concerns (not all) are listed below.

**R: We thank the reviewer for the positive evaluation of the manuscript. We understand**
**that the reviewer might feel like our findings are simple and preliminary. However,**
**please note that with this manuscript, we presented firsthand evidence of differential**
**regulation of callus formation by BES1 and BZR1. We support this by providing**
**mechanistic evidence connecting callus formation to BZR1's direct regulation of the**
**pluripotency master transcription factors ARF7 and ARF19. We further substantiate**
**our findings with genetic evidence showing BZR1's callus and root phenotypes are**
**completely dependent on ARF7 ARF19 module. We think that the combination of**
**molecular and cell biology, phenotyping and genetics establish a well-founded and**
**mechanistic story regarding the differential roles of BZR1 and BES1 in regulating**
**pluripotency.**

**We would like to highlight that, considering the goals of the peer-review process and**
**the efforts of platforms like Review Commons to reduce submission burden and the**
**need for multiple review rounds, a comment such as 'The main concerns (not all) are**
**listed below' may create some ambiguity and potentially lead to additional rounds of**
**review. We believe this could be counterproductive to the streamlined approach that**
**Review Commons aims to support.**

1. BZR1s are the most important transcription factors transcriptional regulating BR-responsive
genes expression. So, it is only BZR1 and BES1 involving in pluripotency acquisition and
callus development? How about other BR mutant including deficient (det2, cpd, dwf4?) and

insensitive mutants (bri1, bin2, bsk?)? Is it BR or BZR1s involving in pluripotency acquisition
and callus development?

**R: This is an interesting question and is similar in nature to what was requested by**
**reviewer1. Please see our reply to reviewer 1 regarding this issue.**

2. The author mentioned in both abstract and results (Figure 1) that bZR1-D and BES1-D
exhibit different phenotypes, which is very interesting. Both bZR1-D and BES1-D can cause an
increased BR signal in plants, but their phenotypes on CIM are completely different. The
authors try to explain that BZR1 expresses in callus but BES1 not. Does the expression
difference between these two genes determine their different function in callus
development? Besides, the expression pattern reflected by marker lines in Figure don't
illustrate significant differences (semi-quantitative statistical analysis is needed). The staining
of cell wall is helpful to see the difference between BES1 and BZR1. Furthermore, the
developmental stages are not the same in BES1 and BZR1 lines in Figure 3C.

**R: We understand the nature of this comment, but we notice that the effect of CIM on**
**the expression levels of both TF is diametrically opposite, therefore no quantification**
**should be needed. We have nonetheless performed this quantification (Figure 4) and**
**can show that BZR1 is significantly induced by CIM treatment while BES1 is not. We**
**do not understand the comment regarding cell wall staining as we have not used any**
**cell wall staining to prevent potential “bleeding effects” of such staining to YFP. We**
**are unsure about what the reviewer means by “the developmental stages” are not the**
**same. Seeds for this experiment were germinated simultaneously and were 7 days old**
**at the moment of culturing in CIM. Then, pictures were taken after tissue was cultured**
**for 24 days (21 days on CIM and 3 days on SIM). By all indications, the callus masses**
**are of the same “developmental age”. It should be also noted that the aim of those**
**pictures was to show that contrary to what had been seen for CIM, BES1 is broadly**
**expressed on calli masses cultured in SIM, and potential differences in development**
**would be irrelevant in that context. Still, we have changed the pictures to have a more**
**similar morphological structure, and we still see the same pattern of expression. We**
**hope this will quell any doubts regarding the issue at hands.**

3. Dose bZR1-1D and BES1-1D has different BR level or auxin level? After CIM, did BR or
auxin level change in Col, bZR1-1D and BES1-D?

**R: We are unsure about the rationale behind this question as it is not phrased in the**
**context of our story. Several studies have indicated that auxin biosynthesis is under a**
**negative feedback loop (e.g. Takato et al 2017 Bios. Biotech and Biochem; Wang et al**
**2020, Nat. Comm; Lee et al 2024 Plant comm.) and thus the massive dosage of 2,4-D**
**in CIM severely impacts endogenous levels of auxin. Concerning BR, our data with**
**BRZ indicates that BR synthesis can be bypassed by both mutants, which is in line**
**with what is known for both mutants in normal developmental conditions. Moreover,**
**given that we demonstrate that BZR1, but not BES1, directly and positively regulates**
**ARF7 and ARF19 in CIM (Figure 5), putative differences in hormones in these mutants**
**cannot be the primary cause for their callus phenotypes.**

4. The authors illustrated that bZR1c and BES1-ko has not phenotypes. BZR family has 6
member and there are published mutant of bZRs. It is necessary to check its phenotype to
gain clear conclusion.

**R: This comment is similar to that raised by reviewer 1. Please see our answer above**
**to this issue.**

5. It is not surprising that BZR1 can directly bind to ARF7/19, as previously reported. Have
the authors detected genes related to other callus attributes, such as WOX5, PLT, SCR, etc?

The authors have BZR1 tag line, which could be used to do ChIP-seq of BZR1 to identified
more downstream targets. Combining with RNA-seq, the big picture and detailed mechanism
could be discovered.

**R: It is correct that it has been previously reported that BZR1 regulates ARF7,**
**however this was shown for hypocotyl tissue and given the often tissue specific**
**nature of signalling networks, we believe our data showing that BZR1 control ARF7**
**during callus formation is very relevant. Perhaps more importantly, to our knowledge**
**we show here for the first time that ARF19 is also under BZR1 regulation. While ARF7**
**and ARF19 work redundantly, they are transcriptionally regulated at different levels**
**(e.g. Moreno -Risueno et al 2010 Science; Orosa-Puente et al 2018 Science). Thus,**
**finding that BZR1 regulates both ARFs transcriptionally is novel and quite relevant to**
**the field.**

**We also notice that our genetic data shows that BZR1's effect on callus is completely**
**dependent on ARF7 and ARF19 and thus other targets that BZR1 might regulate are**
**upstream of ARF7 and ARF19 and cannot explain the callus phenotype of BZR1**
**outside of the direct regulation of the 2 ARFs. We do understand how presenting**
**other targets could be interesting for a general audience. Initially, we performed ChIP-**
**seq of callus masses but unfortunately, we could only perform 1 biological replicate**
**and thus we did not include that data in the manuscript. However, we did use the**
**ChIP-seq data to guide us in testing potential BZR1 targets and we have included**
**some of the ones we were able to reproduce, among them *Scarecrow*, which was also**
**a known target of BZR1 (Tian et al 2021 New Phyt.) and RGF1, which is a completely**
**new target. We iterate that despite reporting these novel targets, callus formation**
**phenotype of *bzr1-D* is completely dependent on ARF7 and ARF19 so they must**
**operate upstream of ARFs.**

6. The authors organize the regulation of lateral organ development and callus development.
It is not complete same mechanism between these two things.

**R: We agree with the reviewer that the processes are not completely the same. When**
**we refer to the processes being similar, is mainly related to their dependency on ARF7**
**and ARF19 signalling.**

7. What is the mechanism of increased *bzr1-1D* and *bes1-D* shoot in SIM (Figure 2D)? What
are the downstream target genes that they regulate shoot growth?

**R: While this question is very interesting, the story we wish to report here clearly**
**focuses on callus formation. Therefore, potential BES1/BZR1 targets during shoot**
**formation are better addressed in subsequent studies.**

Dear Dr. Rodriguez,

Thank you for the transfer of your revised manuscript from Review Commons to our editorial offices. As you know, both original referees have declined to look through the revised manuscript or were completely ignorant to our invitations to re-assess the manuscript. I have now received the report an arbitrator that I asked to re-evaluate your study, you will find below. As you will see, although the arbitrator points out limitations of the study, s/he also states that the study is experimentally solid, and that the referee comments have been adequately addressed.

I will thus proceed with the manuscript that now needs formatting according to our journal style, which I ask you to do in a final revised manuscript. Please carefully review the instructions that follow below.

When submitting your final revised manuscript, we will require:

1) a .docx formatted version of the final manuscript text (including legends for main figures, EV figures and tables), but without the figures included. Figure legends should be compiled at the end of the manuscript text.

2) individual production quality figure files as .eps, .tif, .jpg (one file per figure), of main figures and EV figures. Please upload these as separate, individual files upon re-submission.

The Expanded View format, which will be displayed in the main HTML of the paper in a collapsible format, has replaced the Supplementary information. You can submit up to 5 images as Expanded View. I would thus suggest combining the present supplementary figures to have 5 final figure files. Please follow the nomenclature Figure EV1, Figure EV2 etc. The figure legend for these should be included in the main manuscript document file in a section called Expanded View Figure Legends after the main Figure Legends section. Additional Supplementary material should be supplied as a single pdf file labeled Appendix. The Appendix should have page numbers and needs to include a table of content on the first page (with page numbers) and legends for all content. Please follow the nomenclature Appendix Figure Sx, Appendix Table Sx etc. throughout the text, and also label the figures and tables according to this nomenclature.

3) a complete author checklist, which you can download from our author guidelines (<https://www.embopress.org/page/journal/14693178/authorguide>). Please insert page numbers in the checklist to indicate where the requested information can be found in the manuscript. The completed author checklist will also be part of the RPF.

4) that primary datasets produced in this study (e.g. RNA-seq, ChIP-seq, structural and array data) are deposited in an appropriate public database. If no primary datasets have been deposited, please also state this in a dedicated section (e.g. 'No primary datasets have been generated and deposited'), see below.

The accession numbers and database should be listed in a formal "Data Availability" section (placed after Materials & Methods) that follows the model below. This is now mandatory (like the COI statement). Please note that the Data Availability Section is restricted to new primary data that are part of this study. This section is mandatory. As indicated above, if no primary datasets have been deposited, please state this in this section

Data availability

5) We now request the publication of original source data with the aim of making primary data more accessible and transparent to the reader. Our source data coordinator will contact you to discuss which figure panels we would need source data for and will also provide you with helpful tips on how to upload and organize the files.

6) Our journal encourages inclusion of *data citations in the reference list* to directly cite datasets that were re-used and obtained from public databases. Data citations in the article text are distinct from normal bibliographical citations and should directly link to the database records from which the data can be accessed. In the main text, data citations are formatted as follows: "Data ref: Smith et al, 2001" or "Data ref: NCBI Sequence Read Archive PRJNA342805, 2017". In the Reference list, data citations must be labeled with "[DATASET]". A data reference must provide the database name, accession number/identifiers and a resolvable link to the landing page from which the data can be accessed at the end of the reference. Further instructions are available at: <http://www.embopress.org/page/journal/14693178/authorguide#referencesformat>

7) Regarding data quantification and statistics, please make sure that the number "n" for how many independent experiments were performed, their nature (biological versus technical replicates), the bars and error bars (e.g. SEM, SD) and the test used to calculate p-values is indicated in the respective figure legends (also for potential EV and Appendix figures). Please also check that all the p-values are explained in the legend, and that these fit to those shown in the figure. Please provide statistical testing where applicable. Please avoid the phrase 'independent experiment', but clearly state if these were biological or technical replicates. Please also indicate (e.g. with n.s.) if testing was performed, but the differences are not significant. In case n=2, please show the data as separate datapoints without error bars and statistics. See also: <http://www.embopress.org/page/journal/14693178/authorguide#statisticalanalysis>

Please add to each legend (main and EV figures) a 'Data Information' section explaining the statistics used or providing information regarding replicates and scales.

8) Please add scale bars of similar style and thickness to microscopic images, using clearly visible black or white bars (depending on the background). Please place these in the lower right corner of the images themselves. Please do not write on or near the bars in the image but define the size in the respective figure legend.

9) Please also note our reference format:

10) We updated our journal's competing interests policy in January 2022 and request authors to consider both actual and perceived competing interests. Please review the policy <https://www.embopress.org/competing-interests> and update your competing interests if necessary. Please name this section 'Disclosure and Competing Interests Statement' and put it after the Acknowledgements section.

11) We now use CRediT to specify the contributions of each author in the journal submission system. CRediT replaces the author contribution section. Please use the free text box to provide more detailed descriptions and do NOT add an author contributions section to the manuscript text file. See also guide to authors:

<https://www.embopress.org/page/journal/14693178/authorguide#authorshippinguidelines>

12) Please add up to 5 keywords to the manuscript and order the manuscript sections like this, using these names:

Title page - Abstract - Keywords - Introduction - Results - Discussion - Methods - Data availability section - Acknowledgements - Disclosure and Competing Interests Statement - References - Figure legends - Expanded View Figure legends

13) All Materials and Methods need to be described in the main text using our 'Structured Methods' format, which is required for all research articles. According to this format, the Materials and Methods section should include a Reagents and Tools Table (listing key reagents, primers used, experimental models, software, and relevant equipment and including their sources and relevant identifiers), uploaded as separate file, followed by a Methods and Protocols section in which we encourage the authors

to describe their methods using a step-by-step protocol format with bullet points, to facilitate the adoption of the methodologies across labs. More information on how to adhere to this format as well as downloadable templates (.doc) for the Reagents and Tools Table can be found in our author guidelines (section 'Structured Methods'):

14) Please enter all the funding information also into our submission system during resubmission and make sure this is complete and similar to the one mentioned in the acknowledgements section of the manuscript text file.

In addition, I would need from you:

- a short, two-sentence summary of the manuscript (not more than 35 words).
- three to four short (!) one sentence bullet points highlighting the key findings of your study.
- a schematic summary figure (synopsis image) in jpeg or tiff format with the exact width of 550 pixels and a height of not more than 400 pixels that can be used as a visual synopsis on our website.

I look forward to seeing the final revised version of your manuscript when it is ready. Please let me know if you have questions or comments regarding the revision.

Yours sincerely,

Arbitrator:

The manuscript reveals the distinct functions of two key components of the brassinosteroid signaling pathway, the transcription factors BZR1 and BES1, during callus formation. The study demonstrates that BZR1 expression is upregulated by callus induction medium (CIM), and its gain-of-function mutation leads to increased callus biomass. Using molecular biology and genetic approaches, the authors show that BZR1 functions as a direct transcriptional regulator of ARF7 and ARF9, whose activity downstream of BZR1 is essential for callus formation. Conversely, BES1 plays an opposing role, as its gain-of-function suppresses callus formation on CIM. Expression analyses indicate that BES1 is down-regulated during callus formation. Despite these contrasting roles in callus formation, both BZR1 and BES1 are shown to promote shoot regeneration, underscoring their context-dependent functions in plant development.

The presented study convincingly demonstrates the distinct roles of BZR1 and BES1 in callus formation as well as essential function of BZR1-ARF7/ARF19 regulatory module in callus formation. I find the study experimentally solid, and the authors have adequately addressed the reviewers' comments.

Overall, the study offers interesting initial observations regarding the distinct functions of BZR1 and BES1 in the regulation of callus formation. However, at the molecular level, the work largely focuses on testing previously established molecular links within a different developmental framework, without substantially advancing mechanistic insights. Notably, the mechanisms through which BES1 acts as a negative regulator of callus formation remain unexplored. A deeper investigation into these molecular mechanisms would have provided a more comprehensive understanding of the interplay between these two components of the brassinosteroid signaling pathway. While this work is a solid contribution to the field, shedding light on the functional diversification of transcription factors in brassinosteroid signaling, the molecular insights into their specific roles remain insufficiently developed.

All editorial and formatting issues were resolved by the authors.

Dear Dr. Rodriguez,

Thank you for the submission of your revised manuscript to our editorial offices. Before we can proceed with formal acceptance, I have these few editorial requests I ask you to address in a final revised manuscript:

- Please remove the running title from the manuscript title page.
- Please change the author list in the manuscript text file to first name last name (like in the submission system).
- Please order the sections like this, using these names:
Title page - Abstract - Keywords - Introduction - Results - Discussion - Methods - Data availability section - Acknowledgements (including the funding information) - Disclosure and Competing Interests Statement - References - Figure legends - Expanded View Figure legends
- The data availability section (DAS) is restricted to information on externally deposited datasets. If the Bioline submission is the only such case, please remove the other text from this section, and add a direct link for the Bioline submission. Please make sure this dataset is public latest upon online publication of the paper.
- Please add the corresponding author name(s) to the author checklist.
- Additional Supplementary material should be supplied as a single pdf file labeled 'Appendix'. Please do that for the 'Supplemental Material' you have uploaded. The Appendix should have page numbers and needs to include a table of content on the first page (with page numbers) and legends for all content. Please follow the nomenclature Appendix Figure Sx, Appendix Table Sx etc. throughout the text, and also label the figures and tables according to this nomenclature. Please also add callouts for all Appendix items to the manuscript text file.
- Please make sure that all figure panels (main, EV and Appendix figures) are called out separately and sequentially. Presently, there seems to be no separate callout for panel 3A. A panel 5F is called out, the figure has no panel F. Please check. Moreover, please add callouts for all the Appendix items.
- Please check again that the number "n" for how many independent experiments were performed, their nature (biological versus technical replicates), the bars and error bars (e.g. SEM, SD) and the test used to calculate p-values is indicated in the respective figure legends. Please also check that all the p-values are explained in the legend, and that these fit to those shown in the figure. Please provide statistical testing where applicable. Please avoid the phrase 'independent experiment', but clearly state if these were biological or technical replicates. Please also indicate (e.g. with n.s.) if testing was performed, but the differences are not significant. In case $n=2$, please show the data as separate datapoints without error bars and statistics. See also:
<http://www.embopress.org/page/journal/14693178/authorguide#statisticalanalysis>
- If $n < 5$, please show single datapoints for diagrams. Moreover:
 - Please provide the exact p values in the legends of figures 1B, C, D; 2B, C, D; 3B, 4C-F; 5A, B; 6B, D; EV2 A; S2.
 - Please note that in figures EV3 A, B there is a mismatch between the annotated p values in the figure legend and the annotated p values in the figure file that should be corrected.
 - Please note that the box plots need to be defined in terms of minima, maxima, centre, bounds of box and whiskers, and percentile in the legends of figures 4C-F.
 - Please note that the measure of center for the error bars needs to be defined in the legend of figure EV3 C, S4 A-C;
- Please add to each legend (main, EV and Appendix figures, where applicable) a 'Data Information' section explaining the statistics used or providing information regarding replicates and scales. See:
<https://www.embopress.org/page/journal/14693178/authorguide#figureformat>
- Please remove the instruction text and the sample table from the Reagents and Tools Table and add callouts to the table in the methods section.
- Please add the primer information (Table 1) directly to the Reagents and Tools Table and remove the table from the manuscript files. Please also update the callouts for this table (directing to the Reagents and Tools Table).
- Please add Table 2 to the Appendix. Please place this after the Appendix figures and name it Appendix Table S1. Please add a legend to this table and update the callouts.
- The scale bars in panels 4A and 4B are hardly visible. Please provide images with bigger scale bars.

In addition, I would need from you uploaded separately:

Best,

All editorial and formatting issues were resolved by the authors.

Dr. Eleazar Rodriguez
University of Copenhagen
Biology
Ole Maaloees Vej 5
Copenhagen, orcid||||| 2200
Denmark

Dear Dr. Rodriguez,

I am very pleased to accept your manuscript for publication in the next available issue of EMBO reports. Thank you for your contribution to our journal.

Yours sincerely,

Rev_Com_number: RC-2024-02436
New_manu_number: EMBOR-2024-60767V3
Corr_author: Rodriguez
Title: BZR1 promotes pluripotency and callus formation through direct regulation of ARF7 & ARF19